# Automated Formal Proofs of Combinatorial Identities via Wilf–Zeilberger Guidance and LLMs

Beibei Xiong [1]  Hangyu Lv [1]  Junqi Liu [2]  Yisen Wang [2]  Shaoshi Chen [2]  Jianlin Wang [3]  Zhengfeng Yang [1]
Lihong Zhi [2]

## Abstract

Automating formal proofs of combinatorial identities is challenging for LLM-based provers, as long-horizon proof planning is required and unconstrained search quickly explodes. Symbolic methods such as the Wilf–Zeilberger (WZ) method can achieve a mechanized proof of combinatorial identities by constructing special auxiliary functions and demonstrating that they satisfy specific recurrence relations. We propose WZ-LLM, a neuro-symbolic framework that turns WZ proof plans into *executable proof sketches* in Lean 4 and uses an LLM-based prover to discharge the resulting machine-checkable subgoals. We also train a dedicated WZ-Prover via a Lean-kernel-verified bootstrapping loop with expert-verified iteration, followed by DAPO-based refinement. Experiments show that WZ-LLM achieves a 34% proof success rate on LCI-Test (100 classic combinatorial identities), outperforming strong baselines such as DeepSeek-V3 and Goedel-Prover-V2, and delivering consistent gains on CombiBench and PutnamBench-Comb. These results indicate that our framework provides two complementary strengths: improved direct proving for identities beyond the scope of WZ, and substantially higher end-to-end success when WZ sketches guide a specialized prover. Our code is available at: `https://github.com/BeibeiX0/WZ-LLM`.

---
[1] School of Software Engineering, East China Normal University, Shanghai, China [2] State Key Laboratory of Mathematical Sciences, Academy of Mathematics and Systems Science, University of Chinese Academy of Sciences, Beijing 100190, China [3] School of Computer and Information Engineering, Henan University, Kaifeng, China. Correspondence to: Zhengfeng Yang <zfyang@sei.ecnu.edu.cn>.

*Proceedings of the 43rd International Conference on Machine Learning*, Seoul, South Korea. PMLR 306, 2026. Copyright 2026 by the author(s).

## 1. Introduction

Large language models (LLMs) have recently achieved strong performance on formal automated theorem proving (ATP) benchmarks (Xin et al., 2025; Lin et al., 2025; Ren et al., 2025; Wang et al., 2025a; Chen et al., 2025), enabling formal reasoning toward research-level mathematics (Wei et al., 2024; Yu et al., 2025b). However, scaling LLM-based provers to harder mathematical domains remains challenging, largely due to the lack of principled proof planning and the resulting combinatorial explosion in proof search. Combinatorics is widely considered one of the most difficult areas for ATP (Chen et al., 2025; Google DeepMind, 2025), and *combinatorial identities* form a fundamental and ubiquitous class of statements within it. Consequently, automating machine-checkable proofs of combinatorial identities is a key objective for scalable theorem proving, enabling the construction of reusable verified libraries and reducing the human effort required for formal proof development.

We focus on automating formal proofs of *combinatorial identities* in Lean 4 (Moura & Ullrich, 2021), a task that remains a challenge for current LLM-based provers. Such proofs typically require **long-horizon proof planning**: without an executable global route that specifies key intermediate milestones, an LLM-based prover is forced into largely unconstrained exploration, leading to a combinatorial explosion in proof search. Moreover, **data scarcity** limits the available proof data for this domain. These challenges motivate an approach for automatically generating *executable proof sketches* that decompose a target identity into a sequence of machine-checkable subgoals, thereby constraining the search space and enabling long-horizon reasoning.

The WZ method is a classical and powerful approach to prove combinatorial identities (Zeilberger, 1991; Wilf & Zeilberger, 1990; Gosper, 1978). It provides a natural source of *global proof structure*: a WZ proof is organized around the synthesis of a recurrence together with appropriate boundary and initial conditions, thereby providing a principled *proof plan* for long-horizon reasoning (Zeilberger, 1991). However, turning the WZ method into a fully automated formal proof pipeline remains challenging in practice, due to limited symbolic coverage and the

nontrivial formalization of boundary conditions and obligations in proof assistants. This motivates a sketch-guided neuro-symbolic design: we use the WZ method to *plan and decompose* a target identity into verifiable intermediate goals and leverage LLMs to *discharge* the resulting formal obligations. Meanwhile, LLM-based proving also extends coverage by producing direct Lean proofs when the WZ decomposition is unavailable or fails.

We propose **WZ-LLM**, a framework that translates the WZ proof plan into *executable proof sketches* in Lean—namely, recurrence lemmas and their associated obligations—and leverages a large language model to automatically discharge the resulting subgoals. To make this approach effective in practice and mitigate data scarcity for Lean combinatorics, we develop a Lean-verified training pipeline for WZ-LLM. We start by manually formalizing **307** combinatorial identities from classical textbooks, yielding a high-quality seed corpus for cold-start supervised fine-tuning (SFT). Building on this seed, we expand the data via expert-verified iteration, retaining only *Lean-kernel-verified* model outputs: verified proofs (including WZ-sketch lemmas and direct proofs of WZ-uncovered identities) are added to the training corpus and pruned from the proving-task pool, while unverified attempts are retained for subsequent iterations. We then train a specialized prover, **WZ-Prover**, through cold-start SFT, iterative training on expanded data (Wu et al., 2024), and Dynamic Sampling Policy Optimization (DAPO) refinement (Yu et al., 2025a). Experiments show that WZ-LLM achieves a **34%** end-to-end success rate on a benchmark of **100** classical combinatorial identities in Lean 4, substantially outperforming strong baselines. Furthermore, on *LCI-Test*, WZ-LLM proves **5** identities on which the symbolic-only baseline fails, highlighting complementary coverage beyond purely symbolic methods. WZ-LLM also improves performance on CombiBench and PutnamBench, solving all manually identified identity instances in these benchmarks. Our main contributions are as follows.

- We propose WZ-LLM, a neuro-symbolic framework that converts Wilf–Zeilberger reasoning into *executable proof sketches* in Lean and uses an LLM-based prover to discharge the resulting proof obligations.

- We build a Lean-verified dataset for combinatorial identities and train a domain-specialized prover via a multi-stage pipeline (cold-start SFT, verified bootstrapping, and DAPO refinement), improving long-horizon lemma chaining under distribution shift.

- We evaluate WZ-LLM on a new benchmark *LCI-Test* (100 classical identities) as well as public combinatorics benchmarks, outperforming strong LLM baselines and complementing a symbolic-only baseline by solving additional cases.

## 2. Related Work

**Automated Theorem Proving.** LLM-based automated theorem proving has advanced rapidly in interactive proof assistants such as Lean (Moura & Ullrich, 2021) and Isabelle (Paulson, 1994). One line focuses on step-wise tactic generation with explicit search (e.g., BFS/MCTS) and frequent verifier interaction, achieving strong performance but incurring substantial branching and compute costs (Polu & Sutskever, 2020; Yang et al., 2024; Xin et al., 2025; Wu et al., 2024). Another line emphasizes whole-proof generation to preserve long-range coherence, but may underutilize intermediate verifier signals for systematic correction (Wang et al., 2024; Lin et al., 2025). To mitigate data scarcity and improve long-horizon reasoning, recent methods leverage verifier-filtered bootstrapping and reinforcement learning in verifiable environments (Shao et al., 2024; Ospanov et al., 2025; Xin et al., 2024a). In particular, MA-LoT (Wang et al., 2025b), GAR (Wang et al., 2025c) and Spark-Prover-X1 (Zhou et al., 2025) further improve proving via correction/refinement and scalable verifier-guided training. Our work targets a domain where successful proving critically depends on *explicit proof planning* and studies how symbolic decompositions can be compiled into executable proof sketches to constrain proof search.

**Formal Proof of Combinatorial Identities.** Symbolic computation has long played a central role in proving combinatorial identities by transforming summations into algebraic forms that are amenable to systematic verification. Classical *generating-function* techniques reduce identities to functional equations and coefficient extraction, and remain a standard paradigm in combinatorics (Stanley, 1986; Flajolet & Sedgewick, 2009; Wilf, 2005). Beyond generating functions, elimination-style approaches (e.g., Gröbner-basis methods) establish polynomial or rational identities via ideal membership and symbolic elimination (Cox et al., 1997; Sturmfels, 2002). For hypergeometric and holonomic summations, certificate-based methods such as the Wilf–Zeilberger algorithm and creative telescoping derive recurrences together with rational certificates/invariants whose validity can be mechanically checked (Wilf & Zeilberger, 1992; Petkovsek et al., 1996; Kauers & Paule, 2011). However, while these techniques are highly effective in computer algebra frameworks, their outputs do not directly translate into interactive proof assistants: end-to-end formalization typically requires explicitly reconstructing telescoping arguments, boundary conditions, normalization steps, and non-vanishing side conditions inside the logic, which often dominates the verification cost. Harrison (Harrison, 2015) calculated rational function certificates using the Maxima external computer algebra framework, and subsequently used HOL Light (Harrison, 1996) to rigorously verify that these certificates satisfy the WZ equation (2). Recent efforts have begun to formalize this domain in Lean by construct-

ing dedicated datasets and benchmarks for combinatorial identities (Xiong et al., 2025). In this work, we leverage the WZ method as a decomposition tool that compiles identities into explicit proof obligations, with all proofs verified by Lean 4.

## 3. Preliminaries

We briefly review the Wilf–Zeilberger method, a certificate-based symbolic framework to prove or discover definite summation identities (Wilf & Zeilberger, 1990). Let $F(n, k)$ be a bivariate hypergeometric term, i.e., $F(n+1, k)/F(n, k)$ and $F(n, k+1)/F(n, k)$ are both rational functions of $n$ and $k$. Assume that for each fixed $n$ as a nonnegative integer, $F(n, k)$ vanishes for all but finitely many values of $k$. The goal of the WZ method is to establish identities of the form

$$S(n) := \sum_k F(n, k) = C, \tag{1}$$

where the index $k$ runs on all integers. Note that, for summations of the form $\sum_k f(n, k) = r(n)$ with $r(n) \neq 0$, one can normalize the identity by defining $F(n, k) = f(n, k)/r(n)$, thereby reducing it to the standard form $\sum_k F(n, k) = 1$. This normalization step is crucial in practice, as it simplifies the proof to a uniform goal of showing that the sum equals a constant and thus facilitates systematic certificate construction.

The core of the WZ method is to build a WZ-pair. Specifically, the algorithm seeks an auxiliary term $G(n, k)$ such that the following WZ equation holds:

$$F(n+1, k) - F(n, k) = G(n, k+1) - G(n, k). \tag{2}$$

Under the appropriate boundary conditions, summing both sides of the equation over $k$ makes the right-hand side telescope to zero, yielding

$$S(n+1) - S(n) = 0.$$

This implies that $S(n)$ is independent of $n$, so the identity (1) remains and the constant C can be determined from an initial value $S(n_0)$. It is required that $G(n, k)$ take the multiplicative form $G(n, k) = R(n, k)F(n, k)$, where $R(n, k)$ is a rational function, known as the "WZ certificate". Such certificates can be found automatically via Gosper's algorithm or Zeilberger's creative telescoping.

In this work, we use the WZ method as a principled symbolic certification tool to automatically generate executable proof sketches for combinatorial identities. The resulting sketch transforms the original identity into a set of structured proof obligations, primarily consisting of boundary-condition verification and recurrence-based lemmas. This provides crucial, verifiable structural guidance that constrains the subsequent LLM-based formal proving process and enables scalable proof search in Lean 4.

---

Consider the classical binomial-sum identity

$$\sum_{k=0}^{n} \binom{n}{k}^2 = \binom{2n}{n}. \tag{3}$$

**Step 1 (Normalization).** Divide both sides by the right-hand side and reduce to the standard form

$$\sum_{k=0}^{n} F(n, k) = 1, \qquad F(n, k) := \frac{\binom{n}{k}^2}{\binom{2n}{n}}. \tag{4}$$

**Step 2 (Certificate synthesis).** A symbolic engine synthesizes a rational certificate

$$R(n, k) = -\frac{k^2 (3n - 2k + 3)}{2 (n - k + 1)^2 (2n + 1)}, \tag{5}$$

and defines $G(n, k) := R(n, k) F(n, k)$ such that the WZ relation in Eq. (2) holds.

**Step 3 (Lean-checkable obligations).** The proof of the identity is reduced to a small set of structured obligations: (i) algebraically verify Eq. (2); (ii) show that the boundary terms vanish, i.e., $G(n, 0) = G(n, n+1) = 0$; and (iii) check the base case to conclude that the constant equals 1. Once (i)–(iii) hold, summing Eq. (2) over $k$ yields $S(n+1) - S(n) = 0$ and thus Eq. (3) follows.

---

The boxed derivation exposes an *executable proof sketch*: a recurrence/telescoping relation plus *boundary obligations*, which are exactly the two main types of formal proof task used in our framework.

## 4. Methodology

In this section, we present **WZ-LLM**, a neuro-symbolic framework for formally proving combinatorial identities in Lean 4. As shown in Figure 1, for identities that are amenable to the WZ method, WZ-LLM first constructs an executable **WZ proof sketch** and compiles the original goal into a set of structured proving tasks, including boundary-condition obligations and recurrence-based lemmas. Identities beyond the coverage of the WZ are also incorporated into the task set as direct proving targets. All tasks are then discharged by our specialized prover model, **WZ-Prover**, and the resulting proofs are strictly verified by the Lean kernel. To improve robustness on sketch-induced subgoals and generalization to out-of-distribution identities, we build a broad-coverage formal training corpus with difficulty stratification, and train WZ-Prover via a combination of expert-in-the-loop iterative data expansion, and DAPO refinement with difficulty-smoothing. In general, **WZ-LLM** consists of two stages: *(i) Symbolic Decomposition* using

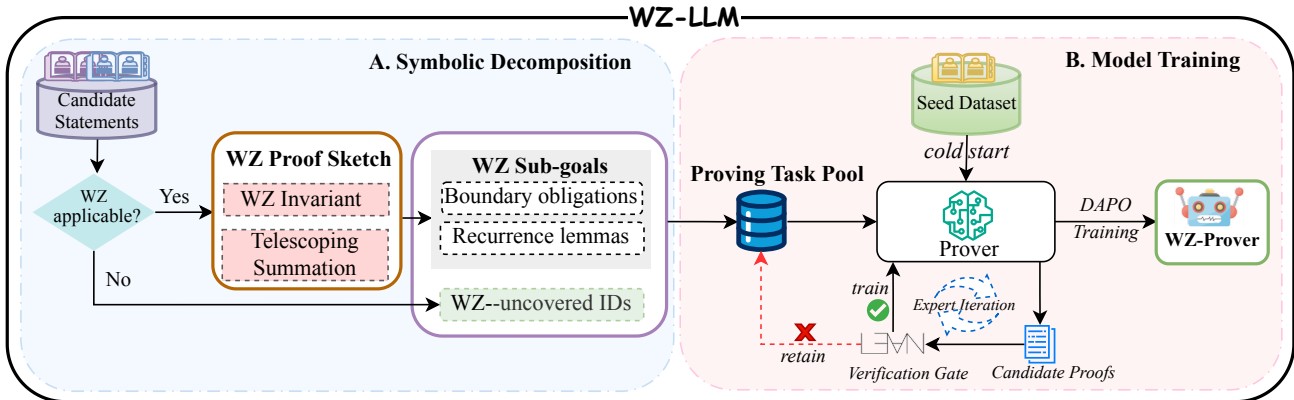

*Figure 1.* **Overview of WZ-LLM. (A) Symbolic decomposition.** Given combinatorial-identity statements, the symbolic engine checks WZ applicability and, when applicable, constructs a WZ proof sketch that is decomposed into sub-goals. Statements that are not amenable to the WZ method are labeled as *WZ-uncovered* targets. All sub-goals and uncovered targets form a shared pool of proving tasks. **(B) Model training.** We cold-start a prover with a seed dataset, iteratively generate proofs for tasks in the pool, and verify them. Lean-verified proofs are retained and used for prune/train updates, while unverified ones are discarded. The expert model is further refined with DAPO to yield **WZ-Prover** for formally discharging both WZ-derived obligations and WZ-uncovered identities.

the WZ method (Section 4.1) and *(ii) Model Training* for WZ-Prover (Section 4.2).

### 4.1. Symbolic Decomposition

The input to WZ-LLM is a formally specified Lean 4 theorem statement, not a natural-language problem description. Given such a Lean statement of a combinatorial identity, WZ-LLM constructs a WZ-style proof sketch and compiles it into a finite set of Lean proof obligations (Figure 1.A). Crucially, it not only produces a WZ certificate/recurrence but also generates the side conditions and case splits needed for successful formalization under Lean's semantics.

**Range Standardization.** The input is modeled into a canonical summation form (e.g. `Finset.range (n+1)` with index starting at 0) using standard transformations: (i) shifting summation ranges not starting at 0; (ii) rewriting bounded sums (`Icc`, `Ico`) into `range-style` sums when possible; (iii) algebraic normalization of hypergeometric factors (factorials/binomials/powers) to reduce syntactic variance. For piecewise specifications (e.g., parity constraints), we compile the goal into structured case splits (e.g. $n$ even/odd) and treat each branch as an independent proving task.

**WZ proof sketch construction.** When the goal is covered by the WZ method, a symbolic engine proposes a recurrence/certificate (e.g., a rational function $R(n, k)$ with $G(n, k) = R(n, k)F(n, k)$). The sketcher then instantiates the telescoping relation and reduces the original identity to standard WZ obligations: (i) a recurrence lemma showing that the normalized sum $S(n)$ is invariant (or satisfies a linear recurrence) and (ii) boundary obligations ensuring that the telescoping sum collapses to boundary terms. The

resulting sketch is an executable Lean skeleton automatically generated by the symbolic stage, rather than a prompt template or an LLM output. Concretely, the CAS-produced certificate $R(n, k)$, the normalized summand $F(n, k)$, and the companion term $G(n, k) = R(n, k)F(n, k)$ are translated into Lean definitions; the WZ equation is then emitted as a local recurrence obligation, and the finite summation range produces boundary and base-case obligations. Thus the WZ-to-Lean translation preserves the symbolic proof plan while exposing only kernel-checkable goals to the neural prover.

**Constraint inference and side-condition generation.** A major practical obstacle in Lean formalization is that algebraic steps (e.g., `field_simp`) require explicit non-vanishing assumptions for denominators and normalization factors. We therefore automatically infer *domain constraints* for each obligation, including when summands or normalizers become zero/undefined, and compile them into Lean side-condition goals. Concretely, we use symbolic simplification and constraint solving to identify problematic points (e.g. zero denominators at boundary indices, negative arguments to factorial-like terms, or sign-sensitive rewrites), and then (i) generate corresponding nonzero lemmas (e.g. $\forall n, k, A(n, k) \neq 0, \forall n, B(n) \neq 0$), (ii) isolate boundary indices as explicit subgoals, and (iii) trigger case splits when constraints depend on discrete properties such as parity.

**Running example.** For the binomial-square identity from Section 3, WZ-LLM emits Lean definitions for $A(n, k) = \binom{n}{k}^2$, $B(n) = \binom{2n}{n}$, $F(n, k) = A(n, k)/B(n)$, the certificate $R(n, k)$ in Eq. (5), and $G(n, k) = R(n, k)F(n, k)$. A key Lean-specific step is that the sketcher also materializes the side conditions required for these rational manipulations in Lean. For this example, the generated obligations fall

into three categories aligned with the proving-task pool:

- **Non-vanishing side conditions** ($\mathcal{T}_{\text{side}}$): well-definedness goals required by algebraic rewriting, including $\forall n\, k,\ k \leq n \rightarrow (\binom{n}{k} : \mathbb{R}) \neq 0$, $\forall n,\ (\binom{2n}{n} : \mathbb{R}) \neq 0$, product forms such as $B(n)B(n+1) \neq 0$, and certificate-denominator constraints such as $n-k+1 \neq 0$ (under $k \leq n$) and $2n + 1 \neq 0$.

- **CAS-assisted ratio lemmas** ($\mathcal{T}_{\text{rec}}$): closed-form quotient identities precomputed by the symbolic engine, e.g., $A(n, k + 1)/A(n, k)$, $A(n + 1, k)/A(n, k)$, and $B(n + 1)/B(n)$, which reduce the recurrence check to arithmetic verification in Lean.

- **WZ relation and boundary obligations** ($\mathcal{T}_{\text{rec}} \cup \mathcal{T}_{\text{bd}}$): the pointwise WZ equation $F(n+1, k) - F(n, k) = G(n, k+1) - G(n, k)$, boundary conditions $G(n, 0) = G(n, n+1) = 0$, and the base case $S(0) = 1$.

For instance, the emitted Lean skeleton includes the following definitions and representative obligations:

```
1  let A : ℕ → ℕ → ℝ := fun n k =>
       (n.choose k)^2
2  let B : ℕ → ℝ := fun n =>
       (2*n).choose n
3  let F : ℕ → ℕ → ℝ := fun n k => A
       n k / B n
4  let R : ℕ → ℕ → ℝ := fun n k =>
5      -(k^2*(3*n - 2*k + 3)) / (2*(n
       - k + 1)^2*(2*n + 1))
6  let G : ℕ → ℕ → ℝ := fun n k => R
       n k * F n k
7  have ne_zeroB : ∀ n, (B n : ℝ) ≠ 0
       := by
8      -- [WZ-Prover]
9  have WZ_inv : ∀ n k, k ≤ n →
10     F (n+1) k - F n k = G n (k+1) -
       G n k := by
11     -- [WZ-Prover]
12 have boundary : ∀ n, G n 0 = 0 ∧ G
       n (n+1) = 0 := by
13     -- [WZ-Prover]
```

Thus Section 3 gives the mathematical WZ plan, while this stage turns the hidden well-definedness requirements into explicit Lean subgoals. WZ-Prover discharges these local goals; once they are accepted by the Lean kernel, the generated skeleton closes the original theorem. Appendix F gives a complete Lean case study with the same classes of generated obligations.

**Proving Task Pool.** Overall, each identity is compiled into a proving-task set $\mathcal{T} = \mathcal{T}_{\text{rec}} \cup \mathcal{T}_{\text{bd}} \cup \mathcal{T}_{\text{side}} \cup \mathcal{T}_{\text{norm}} \cup \mathcal{T}_{\text{case}}$, where $\mathcal{T}_{\text{rec}}$ contains recurrence lemmas, $\mathcal{T}_{\text{bd}}$ contains boundary obligations, $\mathcal{T}_{\text{side}}$ contains non-vanishing and well-typedness conditions required by algebraic tactics, $\mathcal{T}_{\text{norm}}$

contains normalization/range-shift lemmas, and $\mathcal{T}_{\text{case}}$ contains case-split branches. All obligations are expressed as Lean goals and are sent to **WZ-Prover** for formal discharge.

## 4.2. Model Training Pipeline for WZ-Prover

The symbolic stage in Section 4.1 compiles a target identity into a finite set of Lean-checkable proving tasks, including boundary-condition obligations, lemmas based on recurrence, and direct identity goals. If a valid symbolic sketch can be synthesized, releasing these obligations yields an end-to-end Lean proof of the original identity. However, symbolic certificate synthesis may fail, and many identities fall outside the WZ-applicable class. To bridge this gap, we train a combinatorics-specialized Lean prover, **WZ-Prover**, that covers both: (i) *sub-goals* produced by symbolic decomposition, and (ii) *WZ-uncovered identities* that must be proved directly without symbolic guidance. Training proceeds in three stages: (1) supervised cold-start (SFT) on Expert seed dataset; (2) expert-in-the-loop bootstrapping that expands the training set with new Lean-verified lemma and identity proofs; and (3) verifier-based reinforcement learning via DAPO to improve exploration and robustness on hard proving tasks.

**Data construction.** We adopt a data-flow strategy that combines a small high-quality seed dataset, a large pool of unlabeled candidate combinatorial identity statements, and expert-in-the-loop iterative expansion, while retaining a **source-disjoint** test set to evaluate cross-source generalization. The resulting training and test sets are named **LCI-training** and **LCI-Test**, respectively. Specifically, our dataset consists of three components: (i) **Expert seed dataset.** Based on a classical reference (Spivey, 2019), we manually formalized 307 combinatorial identities and provided complete Lean proofs, ensuring all samples are verified in Lean without `sorry`. We further applied a symbolic decomposition method to split these identities into CI proving tasks, generating a total of 1200 sub-goals; (ii) **Candidate statements pool.** From two additional classical sources (Quaintance & Gould, 2015; Gould, 2010), we extracted and formalized 1020 identity statements in Lean 4 as candidate training targets; and (iii) **Source-disjoint test set.** We exclusively reserve two classical combinatorial identity collections (Gould, 1972; Shi, 2001) for evaluation, selecting 100 problems as the test set. To further reduce potential data leakage, expressions are canonized to ensure that the canonical forms of test identities do not appear in the candidate pool or training set.

### 4.2.1. EXPERT-IN-THE-LOOP BOOTSTRAPPING

The baseline model is cold-started on a high-quality expert seed dataset (307 identities with complete Lean proofs, plus 1200 framework-extracted lemma proofs) to acquire basic proving capability. WZ-LLM then generates proofs for the

candidate statement pool from this cold-start model via two routes—symbolic decomposition and direct proving—and we retain only samples that compile and verify in Lean (deduplicated before training).

This process yields two types of new training data: (i) intermediate lemma proofs for WZ sub-goals, and (ii) fully formalized identity proofs. In Round 1, we sample 1020 identities; symbolic decomposition produces 7385 lemma obligations, and we retain 5139 verified lemma proofs plus 32 verified end-to-end proofs. In Round 2, rerunning on the remaining goals yields 532 additional lemma proofs and 79 identity proofs, so expert iteration adds 5671 lemma proofs and 111 identity proofs in total. Notably, the intermediate lemmas align with key steps in the symbolic trajectory, while the full identity proofs help preserve and improve end-to-end performance.

### 4.2.2. DAPO with Difficulty-Smoothing

We further refine WZ-PROVER with a reinforcement-learning (RL) stage to improve robustness on long-horizon lemma chaining and hard end-to-end identities. This refinement has two parts: (i) constructing a difficulty-smoothed RL set from the SFT corpus, and (ii) applying Dynamic Sampling Policy Optimization (DAPO) (Yu et al., 2025a), a stable Long-CoT optimizer that mitigates entropy collapse, truncation-induced reward noise, and training instability.

Our supervised corpus combines (a) a high-quality expert seed set and (b) Lean-verified samples from expert-verified iteration over a pool of candidate statements, including both sketch-induced subgoals (symbolic lemmas) and full identity goals. These goal types have markedly different difficulty profiles: symbolic decomposition yields many short and often repetitive obligations, whereas full identities (and hard subgoals) require longer reasoning chains and fail more frequently. To avoid RL overfitting to frequent easy patterns, we run a rollout-based diagnostic pass to estimate a *pass rate* for each goal under the current policy, use it to tag goals into coarse difficulty bins, and filter the extremes by downsampling near-duplicate easy instances and removing near-zero-pass-rate instances that contribute mostly noisy gradients. The resulting RL set preserves diverse mid-to-hard goals while maintaining a smoother difficulty distribution.

We then optimize with DAPO using a rule-based outcome reward from Lean verification together with a soft overlong punishment:

$$R(\pi; G) = R_{\text{out}}(\pi; G) + \lambda_{\text{len}} R_{\text{len}}(\pi), \qquad (6)$$

where $R_{\text{out}}(\pi; G) \in \{+1, -1\}$ indicates whether $\pi$ kernel-verifies $G$, and $R_{\text{len}}$ applies a gradual penalty near the maximum token budget.

## 5. Experiments

We conduct comprehensive experiments on the classical combinatorial-identity benchmark **LCI-Test** and two external benchmarks, **CombiBench** and **PutnamBench-Comb** (the combinatorics subset of PutnamBench), to evaluate the ability of **WZ-LLM** to produce end-to-end formal proofs in Lean 4. Our evaluation focuses on whether, under a fixed sampling budget, the framework can leverage *proof-sketch* guidance to generate longer and more structured proof chains, and improve success rates on identities outside the coverage of symbolic computation methods (WZ-uncovered). Specifically, Section 5.2 reports the main results on LCI-Test, establishing the overall effectiveness and generality of **WZ-LLM**. Section 5.3 further evaluates its generalization to broader combinatorics problems on CombiBench and PutnamBench-Comb. Finally, Section 5.4 presents ablations that isolate the contributions of key training components.

### 5.1. Experiment Setup

#### 5.1.1. Benchmark and Baselines

We evaluate **WZ-LLM** on three benchmarks: our in-domain benchmark **LCI-Test**, and two external benchmarks, **CombiBench** and **PutnamBench-Comb**. **LCI-Test** is a combinatorial-identity benchmark constructed from classical references (Gould, 1972; Shi, 2001), containing 100 identities to assess the ability of WZ-LLM to formally prove combinatorial identities in Lean 4. **CombiBench** is a challenging benchmark for formal combinatorics theorem proving, consisting of 100 combinatorial problems (Liu et al., 2025). From PutnamBench (Tsoukalas et al., 2024), which contains 672 Lean 4-formalized Putnam problems, we extract a combinatorics subset of 36 problems annotated with the `combinatorics` tag, which we refer to as **PutnamBench-Comb**. These external benchmarks evaluate WZ-LLM's generalization to broader combinatorics problems. We compare WZ-LLM against closed-source LLMs, DeepSeek-V3 (Liu et al., 2024) and Gemini-3.1-Pro-Preview, and open-source provers, including Kimina-Prover-Distill (Wang et al., 2025a), DeepSeek-Prover-V1.5-RL (Xin et al., 2024b), DeepSeek-Prover-V2 (Ren et al., 2025), Goedel-Prover-V2 (Lin et al., 2025), MA-LoT (Wang et al., 2025b), and InternLM-2.5-StepProver (Wu et al., 2024).

#### 5.1.2. Implementation Details

For symbolic decomposition, we use SAGEMATH (Stein & Joyner, 2005) as the underlying computer algebra framework (CAS) to carry out symbolic computations in the WZ/creative-telescoping pipeline. In particular, we rely on the hypergeometric summation routines provided by the

*Table 1.* **Main results on LCI-Test.** We evaluate our framework **WZ-LLM**, which combines a symbolic **WZ-Sketch** decomposer with a specialized Lean prover **WZ-Prover** trained from Goedel-Prover-V2. The upper block reports prior baselines under their respective decoding budgets. The *Ours* block is organized into two parts. *(I) Component effects:* WZ-Sketch +Goedel-Prover-V2 matches the Goedel-Prover-V2 baseline, while WZ-Prover improves direct proving over the same pass@32 budget. *(II) Two-route solving:* **WZ-uncovered** counts identities where WZ decomposition is inapplicable and are solved directly by WZ-Prover; **WZ-Sketch+WZ-Prover** counts WZ-applicable identities solved via sketch-induced obligations discharged by WZ-Prover. **WZ-LLM** aggregates both routes and achieves the highest overall success rate (34/100).

| Method | Model size | Sample budget | LCI-Test |
|---|---|---|---|
| DeepSeek-V3 (Liu et al., 2024) | 685B | pass@32 | 1/100 |
| InternLM-2.5-StepProver (Wu et al., 2024) | 7B | $4 \times 32 \times 600$ | 2/100 |
| MA-LoT (Wang et al., 2025b) | 7B | $16 + 8 \times 2$ | 3/100 |
| Kimina-Prover-Distill (Wang et al., 2025a) | 7B | pass@32 | 6/100 |
| DeepSeek-Prover-V2 (Ren et al., 2025) | 7B | pass@32 | 6/100 |
| Goedel-Prover-V2 (Lin et al., 2025) | 8B | pass@32 | 9/100 |
| Gemini-3.1-Pro-Preview | - | pass@32 | 16/100 |
| *Ours* | | | |
| **WZ-Sketch + Goedel-Prover-V2** | 8B | pass@32 | 9/100 |
| **WZ-Prover** | 8B | pass@32 | **12/100** |
| **WZ-uncovered** | 8B | pass@32 | **5/100** |
| **WZ-Sketch + WZ-Prover** | 8B | pass@32 | **29/100** |
| **WZ-LLM** | 8B | pass@32 | **34/100** |

`sage.combinat` module: given a bivariate hypergeometric term $F(n,k)$, the corresponding WZ certificate $C(n,k)$ is synthesized via `F.WZ_certificate(n,k)`, and the recurrence ratios are simplified using `simplify()`.

On all benchmarks, we report end-to-end proof success rates under the pass@32 metric: a run is counted as successful only if the *entire* proof is accepted by the Lean 4 kernel. The total training cost is 16 GPU-days, and the evaluation cost is 9 GPU-days, all conducted on a $4\times$ L40s-48GB GPU cluster. To quantify the contribution of each component, we report two ablation variants: (i) **WZ-Sketch**, which retains WZ proof-sketch guidance while removing our specialized training pipeline; and (ii) **WZ-Prover**, which retains our training pipeline and specialized prover while disabling WZ proof-sketch guidance. These ablations isolate the gains from symbolic sketching and prover specialization, respectively. In the full WZ-LLM framework, goals are solved via two complementary routes. The *sketch-guided* route invokes symbolic WZ decomposition to compile an identity into a set of Lean-checkable lemma obligations, which are then discharged one by one by **WZ-Prover** (reported as **WZ-Sketch + WZ-Prover** in the table). The *direct* route, used when decomposition is inapplicable or fails, attempts an end-to-end Lean proof directly with **WZ-Prover** (reported

as **WZ-uncovered**). Finally, we report **WZ-LLM** as the union of successes from these two routes. Runtime details for the symbolic pipeline are reported in Appendix E.4.

## 5.2. Main Results

Table 1 reports end-to-end performance on the classic combinatorial identity benchmark LCI-Test (100 problems), measured by the number of fully verified Lean4 proofs. Overall, LCI-Test remains challenging for existing provers under comparable sampling budgets. Among open-source provers, InternLM-2.5-StepProver reaches 2/100, Kimina-Prover-Distill and DeepSeek-Prover-V2 both solve 6/100, and Goedel-Prover-V2 provides the strongest open-source baseline at 9/100. Among proprietary models, DeepSeek-V3 solves 1/100, while Gemini-3.1-Pro-Preview achieves 16/100, outperforming all open-source baselines under the reported setting. Nevertheless, our WZ-LLM achieves 34/100, suggesting that structured symbolic decomposition together with prover specialization can be highly effective for this task.

**Component effects.** Building on Goedel-Prover-V2, we analyze WZ-LLM through two complementary views. Adding **WZ-Sketch** on top of the Goedel-Prover-V2 backend does

not improve end-to-end results (9/100), indicating that sketch-only decomposition is insufficient without a stronger lemma prover. In contrast, our trained **WZ-Prover** improves *direct* end-to-end proving to 12/100 under the same pass@32 budget, demonstrating that the proposed specialization pipeline strengthens the model's ability to discharge core algebraic subgoals even without symbolic guidance.

**Two-route solving.** Our full framework WZ-LLM combines (i) a sketch-guided route for WZ-applicable identities, where **WZ-Sketch** decomposes the goal into Lean-checkable obligations that are discharged by **WZ-Prover**, and (ii) a direct route for **WZ-uncovered** identities, where WZ decomposition is inapplicable and **WZ-Prover** attempts full proofs end-to-end. On LCI-Test, the WZ-uncovered route solves 5 problems, while the sketch-guided route solves 29 problems, yielding 34 solved problems in total when aggregated in **WZ-LLM**. These two counts are complementary rather than overlapping: **WZ-uncovered** refers to identities for which WZ decomposition is unavailable, whereas **WZ-Sketch + WZ-Prover** refers to identities solved after symbolic decomposition into Lean subgoals.

This contrast suggests that our framework provides two complementary capabilities. The specialized **WZ-Prover** improves direct proving on problems outside the scope of WZ, while **WZ**-style symbolic decomposition turns complex identities into Lean-checkable subgoals that can lead to higher end-to-end success when paired with a strong lemma prover. By integrating these two routes within a single framework, **WZ-LLM** achieves an overall end-to-end success rate of $34\%$ on LCI-Test.

*Table 2.* Lemma-proving performance on Symbolic Decomposition subproblems about the base model with our WZ-Prover. WZ-Prover achieves substantially higher lemma accuracy and solves more end-to-end problems.

| Model | #Proved | Acc. | #Solved |
|---|---|---|---|
| Goedel-Prover-V2 | 564 / 1,178 | 47.88% | 0 / 46 |
| WZ-Prover | 864 / 1,178 | 73.34% | 29 / 46 |

To understand why sketch-only guidance fails with the untrained backend, we further perform lemma-level diagnostics on the same 1,178 sketch-induced subgoals (Table 2). Goedel-Prover-V2 proves only 564 lemmas (47.88%), which is insufficient to complete any of the 46 sketched problems because all required obligations must be discharged. Replacing it with **WZ-Prover** raises lemma coverage to 73.34% (864/1,178), enabling symbolic decomposition to translate into 29 end-to-end solutions. Overall, these results show that WZ-style decomposition is most effective when paired with a specialized lemma prover, and that combining sketch-guided and direct proving routes yields the strongest performance.

*Table 3.* **Subgoal-level failure modes for WZ-Prover.** The remaining failures are mainly in recurrence, boundary, and singularity obligations.

| Lemma category | Proved/Generated | Failure |
|---|---|---|
| Non-vanishing (linear) | 573 / 632 | 9.34% |
| Non-vanishing (combinatorial) | 130 / 177 | 26.55% |
| Recurrence relations | 104 / 239 | 56.49% |
| Boundary conditions | 47 / 92 | 48.91% |
| Singularities (identity form) | 10 / 38 | 73.68% |
| Total | 864 / 1,178 | 26.66% |

**Failure analysis.** Of the 66 LCI-Test instances not solved by WZ-LLM, 49 are cases where the symbolic front end fails to produce a usable WZ sketch and the direct WZ-Prover fallback also fails; the remaining 17 have a valid sketch but some Lean subgoals remain unsolved. Table 3 further breaks down the unsolved subgoals. The highest failure rates occur in recurrence relations (56.49%), boundary conditions (48.91%), and singularity obligations (73.68%), indicating that the remaining difficulties are concentrated in mathematically substantive obligations rather than in routine side conditions.

### 5.3. Cross-Dataset Evaluation

Table 4 evaluates cross-dataset generalization on CombiBench and PutnamBench-Comb under a fixed pass@32 sampling budget. Baseline provers remain limited on these heterogeneous combinatorial benchmarks: on CombiBench, Kimina-Prover-Distill, DeepSeek-Prover-V2, and Goedel-Prover-V2 solve 6/100, 8/100, and 12/100 problems, respectively; on PutnamBench-Comb, they solve 0/36, 2/36, and 0/36 problems.

**Component effects.** Using **WZ-Sketch** with the baseline Goedel-Prover-V2 provides little benefit: Goedel-Prover-V2 alone solves 12/100 problems on CombiBench, and the sketch-guided variant adds only one extra proof (13/100), while on PutnamBench-Comb it offers no improvement (remaining at 0/36). This indicates that symbolic decomposition alone does not translate into end-to-end success when the backend cannot reliably discharge the induced lemma obligations. In contrast, our trained **WZ-Prover** substantially improves *direct* proving, reaching 15/100 on CombiBench and 3/36 on PutnamBench-Comb, outperforming Goedel-Prover-V2 by +3 and +3 problems, respectively.

**Two-route accounting in WZ-LLM.** We decompose **WZ-LLM**'s performance into its two proof routes. On CombiBench, the **WZ-uncovered** (WZ-inapplicable) route contributes 15 solved problems via direct proving, while the

*Table 4.* **Cross-dataset evaluation (pass@32) on CombiBench and PutnamBench-Comb.** All our variants are built on Goedel-Prover-V2 as the base model. The table separates (i) *component ablations* (*WZ-Sketch+Goedel* vs. *WZ-Prover*) and (ii) *two-route accounting* in **WZ-LLM**: the **WZ-uncovered** route solves goals where WZ decomposition is inapplicable, while the **WZ-Sketch+WZ-Prover** route solves WZ-applicable goals via symbolic decomposition plus lemma discharge. **WZ-LLM** aggregates both routes.

| Method | Model size | Sample budget | CombiBench | PutnamBench-Comb |
|---|---|---|---|---|
| Kimina-Prover-Distill | 8B | pass@32 | 6/100 | 0/36 |
| DeepSeek-Prover-V2 | 7B | pass@32 | 8/100 | 2/36 |
| Goedel-Prover-V2 | 8B | pass@32 | 12/100 | 0/36 |
| *Ours* | | | | |
| **WZ-Sketch + Goedel-Prover-V2** | 8B | pass@32 | 13/100 | 0/36 |
| **WZ-Prover** | 8B | pass@32 | 15/100 | 3/36 |
| **WZ-uncovered** | 8B | pass@32 | 15/100 | 2/36 |
| **WZ-Sketch + WZ-Prover** | 8B | pass@32 | 1/100 | 1/36 |
| **WZ-LLM** | 8B | pass@32 | **16/100** | **3/36** |

sketch-guided route (**WZ-Sketch+WZ-Prover**) contributes 1 additional solved problem through symbolic decomposition and lemma discharge, yielding **16/100** overall. On PutnamBench-Comb, direct proving solves 2/36 problems, and the sketch-guided route adds 1/36, resulting in **3/36**.

Overall, these results reinforce the same conclusion as on LCI-Test: **WZ-Prover** is the main driver of generalization via stronger direct proof capability, while sketch-guided decomposition can provide additive gains when WZ decomposition is applicable and the resulting obligations are within the lemma prover's reach.

### 5.4. Effect of Training Stages

Table 5 reports stage-wise ablations for WZ-PROVER on LCI-Test, isolating the effects of seed SFT, expert-verified iteration, and DAPO refinement. The cold-start model trained on the seed set alone achieves only 1/100 at pass@1, 3/100 at pass@8, and 9/100 at pass@32, indicating that the expert seed supervision is insufficient to generalize to the harder identities in LCI-Test. Incorporating expert-verified iteration improves performance to 3/100, 6/100, and 10/100, consistent with verifier-filtered data growth that introduces additional WZ-induced obligations and diverse identity proofs (see Appendix E). After deduplication and filtering, the resulting SFT corpus contains 418 Lean-verified identities and 6,871 lemma proofs.

Finally, DAPO refinement further raises performance to 4/100, 6/100, and 12/100. Notably, the improvement is concentrated at larger sampling budgets (pass@32), which suggests that RL primarily helps on harder instances that require longer proof trajectories and benefit from increased

exploration, rather than on the easiest problems already solved by SFT. Additional implementation details for the RL set construction (difficulty smoothing via pass-rate filtering) and DAPO settings are provided in Appendix E.

| Training stage | LCI-Test | | |
|---|---|---|---|
| | pass@1 | pass@8 | pass@32 |
| *SFT (seed only)* | 1/100 | 3/100 | 9/100 |
| *+ expert-iteration* | 3/100 | 6/100 | 10/100 |
| *+ DAPO refinement* | 4/100 | 6/100 | 12/100 |

*Table 5.* **Training-stage ablations for WZ-Prover on LCI-Test.**

## 6. Conclusion

We propose WZ-LLM, a neuro-symbolic framework that integrates the Wilf–Zeilberger method with LLM-based proving for combinatorial identities in Lean 4. WZ-LLM translates WZ proof plans into executable proof sketches and uses a trained prover, WZ-Prover, to discharge the resulting machine-checkable subgoals. This approach provides principled long-horizon proof planning while extending coverage beyond purely symbolic methods. Across multiple benchmarks, WZ-LLM achieves higher proof success rates than strong LLM baselines; on LCI-Test, it additionally proves identities that defeat a symbolic-only baseline. Overall, these results indicate that coupling symbolic proof planning with learned formal reasoning is a promising direction for machine-checkable scalable verification in combinatorics.

## Impact Statement

This paper presents a Lean-verified proving framework that combines symbolic WZ decomposition with a trained neural prover. The approach can improve automation for formalizing combinatorial identities and reduce the manual effort needed to obtain kernel-checked proofs. We do not anticipate direct negative societal impacts beyond standard concerns about overreliance on automated tools; all results are validated by the Lean 4 kernel.

## Acknowledgements

This work was supported in part by the National Key Research and Development Program of China under Grant 2023YFA1009402, the Strategic Priority Research Program of Chinese Academy of Sciences under Grant XDA0480501, in part by the NSFC grant (No. 12271511) and the CAS Project for Young Scientists in Basic Research (No. YSBR-034), in part by ECNU Multifunctional Platform for Innovation (001).

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

## A. Limitation

The symbolic component of WZ-LLM is constrained by the scope of the classical WZ method: it is mainly applicable to identities reducible to bivariate hypergeometric summations, and relies on SageMath for certificate synthesis, which may time out on high-degree instances. On the neural side, our trained 8B prover remains sensitive to the training distribution and may struggle on the most demanding subgoals (e.g., complex recurrence and singularity obligations). Incorporating additional symbolic algorithms (e.g., multi-sum creative telescoping, $q$-hypergeometric methods) and replacing or augmenting the local prover with more capable external reasoning models are natural next steps.

## B. Glossary of Symbolic and Formal-Proof Terms

To better understand the terms, we provide this chart that explains in detail every term, abbreviation, and corresponding tool.

### B.1. WZ and Symbolic-Computation Terms

**Hypergeometric term.** A bivariate term $F(n, k)$ is *(bi-)hypergeometric* if the ratios $F(n, k+1)/F(n, k)$ and $F(n+1, k)/F(n, k)$ are rational functions in $(n, k)$. Many WZ/creative-telescoping procedures apply only to identities whose summands admit such rational ratios.

**Telescoping sum.** A sum of the form $\sum_k \big(G(n, k+1) - G(n, k)\big)$ is *telescoping*: all intermediate terms cancel, leaving only boundary contributions. Thus, if the boundary terms vanish, the whole sum is evaluated to $0$.

**WZ pair.** A *WZ pair* is a pair of terms $(F, G)$ that satisfy

$$F(n+1, k) - F(n, k) = G(n, k+1) - G(n, k),$$

so that summing over $k$ yields a (typically constant) behavior of $S(n) = \sum_k F(n, k)$ under suitable boundary conditions.

**WZ certificate** $R(n, k)$**.** In WZ proofs one often sets $G(n, k) = R(n, k)F(n, k)$, where $R(n, k)$ is a rational function. This $R$ is called a *certificate*; once proposed by a CAS, the remaining work is to verify the induced rational identity and the boundary conditions within the proof assistant.

**Creative telescoping / Zeilberger algorithm.** *Creative telescoping* generalizes WZ by producing higher-order relations

$$\sum_{j=0}^{J} a_j(n) \, F(n+j, k) = G(n, k+1) - G(n, k),$$

which induce a linear recurrence for $S(n) = \sum_k F(n, k)$ after summing over $k$. The case $J = 1$ recovers the classical WZ equation.

**Recurrence relation.** A *recurrence* is an equation that relates $S(n)$ to different $n$ (e.g., first-order or higher-order linear recurrences). In our setting, proving the recurrence together with a base case is sufficient to determine $S(n)$ and close the target identity.

**Boundary conditions / boundary obligations.** To justify telescoping in a finite range (e.g., $k = 0, \ldots, n$), one must prove that the boundary terms vanish (or match) at the endpoints, such as $G(n, 0) = 0$ and $G(n, n+1) = 0$. These are emitted as explicit *boundary obligations*.

**Normalization.** For an identity $\sum_k f(n, k) = r(n)$ with $r(n) \neq 0$, we normalize to $\sum_k F(n, k) = 1$ by setting $F(n, k) = f(n, k)/r(n)$. This reduces the goal to proving the constancy of the normalized sum plus a base case.

**Summation range transformation.** Many identities require rewriting sums to a canonical range, e.g.,

$$\sum_{k=a}^{b} f(k) = \sum_{k=0}^{b-a} f(k+a),$$

Separate a few boundary terms. Such transformations are often needed to match the requirements of WZ-style decomposition.

**Singularity / nonzero side conditions.** Symbolic certificates frequently introduce denominators. Formal verification therefore requires *side conditions* ensuring that those denominators are nonzero in the relevant domain (e.g., to justify `field_simp` steps in Lean).

### B.2. Automated Theorem Proving Terms

**Kernel verification.** Lean's trusted kernel checks that the produced proof term/script is valid. All reported successes correspond to proofs accepted by the kernel.

**Proof obligation.** A *proof obligation* is a subgoal generated by a proof sketch (e.g., a boundary condition or a recurrence lemma) that must be proved to complete the overall proof.

**pass@k (e.g., pass@32).** `pass@k` measures whether at least one of the $k$ independently sampled proof attempts verifies. It is standard for stochastic theorem provers where success is probabilistic.

**Executable proof sketch.** In this paper, a *proof sketch* is not a natural-language outline but a structured, machine-checkable decomposition (e.g., recurrence + boundary obligations + base case) that can be executed in Lean as a sequence of formal subgoals.

**WZ-covered vs. WZ-uncovered identities.** An identity is *WZ-covered* if the symbolic engine can synthesize a WZ/creative-telescoping certificate and emit a sketch. Otherwise, it is *WZ-uncovered* and is treated as a direct proving target.

**WZ sub-goals and CI proving tasks.** *WZ sub-goals* refer to the boundary obligations and recurrence lemmas produced by sketch construction. *CI proving tasks* include both these sub-goals and the WZ-uncovered identities, forming the unified task set solved by WZ-Prover.

**Expert iteration / verifier-filtered bootstrapping.** We repeatedly sample candidate proofs from the current prover for a pool of unproven statements, keep only Lean-verified proofs, and add the resulting pairs (statement, proof) back to the training set. We then update the prover and continue on the remaining statements, expanding coverage while maintaining formal correctness.

**Goal / theorem statement.** A *goal* $G$ denotes a Lean proposition to be proved (either a full identity or an intermediate lemma). We write $\mathcal{V}(G, \pi) = \text{true}$ if the Lean kernel verifies that the tactic script $\pi$ proves $G$.

**Candidate statement.** A *candidate statement* is an unlabeled target identity extracted from external sources (e.g., tables of identities) and normalized into a uniform, Lean-parseable form. The pool of candidate statements is used for expert-in-the-loop bootstrapping and data expansion.

## C. Pseudocode for WZ-LLM and WZ-Prover Training

Our approach combines the complementary strengths of symbolic computation and learned proof search: symbolic WZ decomposition externalizes long-horizon proof planning into a set of locally checkable obligations, while a specialized Lean prover efficiently discharges the resulting algebraic and boundary subgoals. We present the inference-time framework WZ-LLM (Algorithm 1) and the training pipeline for WZ-Prover (Algorithm 2). At inference time, WZ-LLM queries a symbolic engine to construct a WZ-style proof sketch, consisting of a recurrence structure together with boundary-condition obligations. When WZ decomposition succeeds, the framework reduces the original identity into a set of Lean-checkable subgoals, which are then discharged by WZ-Prover in Lean4. If the identity is *WZ-inapplicable* (i.e., no certificate/sketch can be synthesized), WZ-LLM falls back to direct proving using WZ-Prover. All generated proofs and proof fragments are strictly verified by the Lean kernel.

**Algorithm 1: Sketch-guided neuro-symbolic proving.** Algorithm 1 describes the inference-time procedure of WZ-LLM. Given a target Lean goal $S$, the framework maintains a queue $\mathcal{Q}$ of pending obligations and iteratively processes each goal $G$ popped from the queue. To avoid redundant work (e.g., repeated lemmas produced by different decompositions), we maintain a visited set $\mathcal{S}_{\text{seen}}$ and skip any goal that has already been processed.

---

**Algorithm 1** WZ-LLM: Sketch-Guided Neuro-Symbolic Proving in Lean4

---

**Input:** target goal $S$, WZ engine $\mathcal{E}$, prover $\mathcal{M}$ (WZ-Prover), Lean verifier $\mathcal{V}$
**Output:** Lean4-verified proof of $S$ or FAIL
$\mathcal{Q} \leftarrow \{S\}$ {queue of pending obligations}
$\mathcal{P} \leftarrow \emptyset$ {kernel-checkable proof fragments}
$\mathcal{S}_{\text{seen}} \leftarrow \emptyset$ {visited goals (dedup)}
**while** $\mathcal{Q} \neq \emptyset$ **do**
  Pop $G$ from $\mathcal{Q}$
  **if** $G \in \mathcal{S}_{\text{seen}}$ **then**
    **continue**
  **end if**
  $\mathcal{S}_{\text{seen}} \leftarrow \mathcal{S}_{\text{seen}} \cup \{G\}$
  (`applicable`, `sketch`, $\mathcal{L}$) $\leftarrow \mathcal{E}(\text{WZ-DECOMPOSE}, G)$
  **if** `applicable` **then**
    $\mathcal{P} \leftarrow \mathcal{P} \cup \{\texttt{sketch}\}$ {Lean skeleton reducing $G$ to obligations in $\mathcal{L}$}
    **for** each $\ell \in \mathcal{L}$ **do**
      **if** $\ell \notin \mathcal{S}_{\text{seen}}$ **then**
        Push $\ell$ into $\mathcal{Q}$
      **end if**
    **end for**
  **else**
    $\pi \leftarrow \mathcal{M}(G)$ {direct proving for WZ-inapplicable goals}
    **if** $\mathcal{V}(G, \pi)$ **then**
      $\mathcal{P} \leftarrow \mathcal{P} \cup \{\pi\}$
    **else**
      **return** FAIL
    **end if**
  **end if**
**end while**
**return** ASSEMBLE($\mathcal{P}$) {assemble fragments; Lean kernel checks final proof}

---

For each goal $G$, WZ-LLM queries the symbolic backend $\mathcal{E}$ via WZ-DECOMPOSE. If the goal is *WZ-applicable*, the backend returns (i) an executable Lean *sketch* and (ii) a finite set of Lean-checkable obligations $\mathcal{L}$, which typically include recurrence/ratio checks, boundary conditions, normalization steps, and non-vanishing side conditions needed to justify telescoping. The sketch is stored as a kernel-checkable fragment in $\mathcal{P}$, while each obligation $\ell \in \mathcal{L}$ is pushed into $\mathcal{Q}$ for subsequent discharge (subject to deduplication by $\mathcal{S}_{\text{seen}}$).

If WZ-DECOMPOSE reports that $G$ is *WZ-inapplicable*, the system falls back to direct proving: the learned prover $\mathcal{M}$ (WZ-Prover) generates a tactic script $\pi \leftarrow \mathcal{M}(G)$, which is accepted only if the Lean verifier $\mathcal{V}$ confirms $\mathcal{V}(G, \pi) = \texttt{true}$. Any verifier failure causes the procedure to return FAIL. Finally, once $\mathcal{Q}$ is exhausted, ASSEMBLE($\mathcal{P}$) combines all verified fragments (sketches and lemma proofs) into a complete end-to-end proof, which is again checked by the Lean kernel.

**Algorithm 2: Training pipeline for WZ-Prover.** Algorithm 2 details the three-stage training pipeline that produces WZ-Prover. The procedure starts from a small expert-verified seed dataset $\mathcal{D}_{\text{seed}}$ and performs supervised fine-tuning (SFT) to obtain an initial prover $\mathcal{M}$. It then performs expert-in-the-loop dataset expansion for $T$ rounds over a pool of candidate goals $\mathcal{S}_{\text{cand}}$. In each round $t$, the algorithm generates new training samples $\mathcal{D}_{\text{new}}$ from two sources. When a sampled goal $S$ is *WZ-applicable*, we decompose it into obligations $\mathcal{L}$ and attempt to prove each lemma $\ell \in \mathcal{L}$ using the current prover $\mathcal{M}$. Draft proofs $\hat{\pi}_\ell$ are then curated through EXPERTREPAIRANDFILTER, which repairs incomplete or partially correct traces, filters out low-quality outputs, and returns a cleaned pair $(\ell', \pi_\ell)$ only when a proof is well-formed and verifiable. When $S$ is *WZ-inapplicable*, we instead attempt to directly prove the full identity with $\mathcal{M}$ and apply the same expert repair/filtering step, yielding $(S', \pi_S)$ when successful. All accepted samples must pass Lean verification under $\mathcal{V}$.

After collecting $\mathcal{D}_{\text{new}}$, we augment the training set $\mathcal{D} \leftarrow \mathcal{D} \cup \mathcal{D}_{\text{new}}$ and retrain the prover with SFT on the expanded Lean-verified dataset, repeating for $T$ rounds. Finally, we apply DAPO refinement to further improve long-horizon lemma

---

**Algorithm 2** WZ-Prover Training: Seed SFT → Expert Iteration → DAPO Refinement

---

  **Input:** seed proofs $\mathcal{D}_{\text{seed}}$, candidate goals $\mathcal{S}_{\text{cand}}$, WZ engine $\mathcal{E}$, Lean verifier $\mathcal{V}$, expert rounds $T$
  **Output:** trained prover $\mathcal{M}^{\star}$ (WZ-Prover)
  $\mathcal{M} \leftarrow \text{SFT}(\mathcal{D}_{\text{seed}})$ {cold-start}
  $\mathcal{D} \leftarrow \mathcal{D}_{\text{seed}}$
  **for** $t = 1$ **to** $T$ **do**
    $\mathcal{D}_{\text{new}} \leftarrow \emptyset$
    **for each** sampled goal $S \in \mathcal{S}_{\text{cand}}$ **do**
      $(\texttt{applicable}, \texttt{sketch}, \mathcal{L}) \leftarrow \mathcal{E}(\text{WZ-DECOMPOSE}, S)$
      **if** $\texttt{applicable}$ **then**
        **for each** lemma $\ell \in \mathcal{L}$ **do**
          $\hat{\pi}_\ell \leftarrow \mathcal{M}(\ell)$
          $(\ell', \pi_\ell) \leftarrow \text{EXPERTREPAIRANDFILTER}(\ell, \hat{\pi}_\ell)$
          **if** $(\ell', \pi_\ell) \neq \emptyset$ **and** $\mathcal{V}(\ell', \pi_\ell)$ **then**
            $\mathcal{D}_{\text{new}} \leftarrow \mathcal{D}_{\text{new}} \cup \{(\ell', \pi_\ell)\}$
          **end if**
        **end for**
      **else**
        $\hat{\pi}_S \leftarrow \mathcal{M}(S)$
        $(S', \pi_S) \leftarrow \text{EXPERTREPAIRANDFILTER}(S, \hat{\pi}_S)$
        **if** $(S', \pi_S) \neq \emptyset$ **and** $\mathcal{V}(S', \pi_S)$ **then**
          $\mathcal{D}_{\text{new}} \leftarrow \mathcal{D}_{\text{new}} \cup \{(S', \pi_S)\}$
        **end if**
      **end if**
    **end for**
    $\mathcal{D} \leftarrow \mathcal{D} \cup \mathcal{D}_{\text{new}}$
    $\mathcal{M} \leftarrow \text{SFT}(\mathcal{D})$ {retrain on expanded data}
  **end for**
  {DAPO refinement with difficulty-smoothing data curation (Section 4.2.2)}
  $\mathcal{M}^{\star} \leftarrow \text{DAPO}(\mathcal{M};\ R(\pi; G))$
  **return** $\mathcal{M}^{\star}$

---

chaining and robustness. In contrast to modifying the reward design, our refinement relies on *difficulty-smoothing data curation*: we run a rollout-based diagnostic pass to estimate goal pass rates under the current policy, bucket goals into coarse difficulty bins, and then downsample redundant easy instances while removing near-zero-pass-rate goals that contribute mostly noisy gradients (Section 4.2.2). DAPO is then applied on this curated RL set using the same reward definition as in Eq. 6. The resulting model $\mathcal{M}^{\star}$ is returned as the final WZ-Prover.

## D. Training Data Details

Our training corpus contains two major categories: (i) *symbolic-induced lemmas* produced by WZ-style decomposition (and related algebraic simplifications), and (ii) *end-to-end identity proofs* for goals that can be discharged directly (including identities not covered by WZ decomposition). All retained samples must compile and verify in Lean; we also deduplicate the proofs before adding them to the training set.

**Expert-verified iteration.** We perform expert-verified dataset expansion for $T = 2$ rounds. In each round, we sample identities from the candidate pool, apply symbolic decomposition to generate lemma obligations, and let the prover attempt both (i) lemma proofs for sketch-induced subgoals and (ii) direct end-to-end identity proofs. We retain only samples that compile and verify in Lean, and we deduplicate accepted proofs before adding them to the training corpus in Table 6.

**RL dataset size and composition.** The RL stage operates on a filtered subset of the Lean-verified SFT corpus obtained from the seed set plus expert-verified iteration (Table 6). To smooth the difficulty mismatch between sketch-induced lemmas (often short and repetitive) and full-identity goals (typically long-horizon and failure-prone), we run a rollout-based diagnostic pass and estimate a pass rate for each goal under the current policy. We then construct the RL set by downsampling large clusters

| Stage | Accepted identities | Seed lemmas | Generated obligations | Accepted lemmas |
|---|---|---|---|---|
| Seed dataset | 307 | 1,200 | – | 1,200 |
| Expert iteration (Round 1) | 32 | – | 7,385 | 5,139 |
| Expert iteration (Round 2) | 79 | – | 2,246 | 532 |
| **Total (accepted)** | **418** | – | – | **6,871** |

*Table 6.* **Training data growth via expert-verified iteration.** We start from a seed corpus of 307 Lean-verified identities and 1,200 framework-extracted lemma proofs. Each iteration samples identity statements, decomposes them into lemma obligations when applicable, and retains only Lean-verified and deduplicated outputs. Across two rounds, expert iteration adds 111 new identity proofs and 5,671 lemma proofs, yielding 418 identities and 6,871 lemmas in total.

of high-pass-rate, near-duplicate easy instances and excluding near-zero-pass-rate goals whose rollouts are dominated by truncation or early failure. The resulting RL set emphasizes diverse mid-to-hard goals, contains a higher fraction of long-horizon proof trajectories, and still mixes lemma-style subgoals (e.g., ratio/non-vanishing obligations) with full identity goals to preserve end-to-end capability. All RL prompts follow the same prover input template, and each rollout is scored by Lean kernel verification plus the soft overlong punishment in Eq. 6.

### D.1. Lemma types and data format

Symbolic-induced training instances are dominated by two recurring lemma families: (i) **ratio lemmas**, which expose CAS-precomputed term ratios (e.g., A(n,k+1)/A(n,k) or A(n+1,k)/A(n,k)) to avoid expensive normalization inside Lean; and (ii) **non-vanishing (well-definedness) lemmas**, which establish that denominators are nonzero, enabling safe use of `field_simp` and preventing singularities during normalization and recurrence checking. All lemma samples are stored as instruction-following triples (`instruction`/`input`/`output`): the `input` contains a Lean goal template, and the `output` includes (a) a brief proof plan and (b) a complete Lean4 proof script. We give one example for each lemma type below (formatted identically to training instances).

The first example illustrates a typical "local obligation" that frequently appears in our pipeline (e.g., when normalizing expressions or discharging boundary side conditions). Such goals are not mathematically deep, but they are essential for making later `simp`/`field_simp` steps well-defined and for keeping the main proof trace stable.

---

**Statement (Lean goal).**

```
1  import Mathlib
2
3  open Nat Finset BigOperators
4
5  theorem factorial_succ_ge_one {n : ℕ} (h : 1 ≤ n) :
6    n * (n - 1)! = n! := by
7    -- to be completed
```

**Proof / Model Output.**
**Problem.** Show that for any natural number $n$ with $1 \leq n$, we have

$$n \cdot (n-1)! = n!.$$

**Key observations.**

1. The hypothesis $1 \leq n$ implies $0 < n$, which lets us rewrite $n$ as $(n-1)+1$ via `Nat.succ_pred_eq_of_pos`.

2. Lean's factorial recurrence is encoded as `Nat.factorial_succ`: `(m+1)! = (m+1) * m!`. Therefore, once we rewrite $n$ as `(n-1)+1`, the goal reduces to a direct rewrite by `Nat.factorial_succ`.

3. No induction is needed: the proof is a one-step normalization that bridges the arithmetic identity `n = (n-1)+1` with the definitional equation of factorial.

---

**Proof plan.**

1. Derive `0 < n` from the assumption `1 ≤ n`.

2. Rewrite `n` as `(n - 1) + 1` using `Nat.succ_pred_eq_of_pos`.

3. Replace `n!` by `((n-1)+1)!` and rewrite it using `Nat.factorial_succ`.

4. Finish by simplification (`simp`) and associativity/commutativity normalization (`ring` is unnecessary for ℕ here).

**Complete Lean 4 Proof.**

```
1  import Mathlib
2
3  open Nat Finset BigOperators
4
5  theorem factorial_succ_ge_one {n : ℕ} (h : 1 ≤ n) :
6    n * (n - 1)! = n! := by
7    have hn : 0 < n := Nat.pos_of_ne_zero (by
8      -- 1 ≤ n implies n ≠ 0
9      exact Nat.ne_of_gt (Nat.lt_of_lt_of_le Nat.zero_lt_one h))
10   -- rewrite n as (n - 1) + 1
11   have hn' : (n - 1).succ = n := Nat.succ_pred_eq_of_pos hn
12   -- turn it into a rewriting form n = (n - 1) + 1
13   have hn'' : n = (n - 1) + 1 := by
14     -- succ is +1 in Nat
15     simpa [Nat.succ_eq_add_one] using hn'.symm
16   -- use factorial_succ on m = n-1
17   -- (n-1+1)! = (n-1+1) * (n-1)!
18   -- then rewrite (n-1+1) as n
19   calc
20     n * (n - 1)! = ((n - 1) + 1) * (n - 1)! := by
21       simpa [hn'']
22     _ = ((n - 1) + 1)! := by
23       simpa [Nat.factorial_succ]
24     _ = n! := by
25       simpa [hn'']
```

The second example illustrates a typical *non-vanishing / well-definedness* obligation produced by symbolic preprocessing: from an arithmetic assumption $n > m$ in ℕ, the prover must show that a real denominator term $(n - m + 1)$ is nonzero.

**Statement (Lean goal).**

```
1  import Mathlib
2
3  open Real Nat Finset BigOperators Polynomial
4
5  set_option maxHeartbeats 8000000000
6  theorem hwz (n m : ℕ) (htotalNumidx : n > m) :
7    ((↑n - ↑m + 1) : ℝ) ≠ 0 := by
8    -- to be completed
```

**Proof / Model Output.**
**Problem.** Given natural numbers $n, m$ with $n > m$, prove the real expression

$$(n - m + 1) \neq 0.$$

**Key observations.**

1. From $n > m$ we get $m < n + 1$ in $\mathbb{N}$ immediately. This is a cleaner arithmetic target than reasoning directly about a casted subtraction in $\mathbb{R}$.

2. If we can show $(m : \mathbb{R}) < (n + 1 : \mathbb{R})$, then

$$(n - m + 1 : \mathbb{R}) = (n + 1 : \mathbb{R}) - (m : \mathbb{R}) > 0,$$

hence it cannot be zero (discharged by `linarith`).

3. The bridge between the $\mathbb{N}$-inequality and the $\mathbb{R}$-inequality is handled by `norm_cast`. The remaining linear arithmetic on naturals is solved by `omega`.

**Proof plan.**

1. Reduce the non-vanishing goal to a positivity/inequality statement: it suffices to prove $(m : \mathbb{R}) < (n + 1 : \mathbb{R})$, then conclude by `linarith`.

2. Use `norm_cast` to turn $(m : \mathbb{R}) < (n + 1 : \mathbb{R})$ into the natural-number inequality $m < n + 1$.

3. Prove $m < n + 1$ from the hypothesis $n > m$ using `omega`.

**Complete Lean 4 Proof.**

```
1  import Mathlib
2
3  open Real Nat Finset BigOperators Polynomial
4
5  set_option maxHeartbeats 8000000000
6  theorem hwz (n m : ℕ) (htotalNumidx : n > m) :
7    ((↑n - ↑m + 1) : ℝ) ≠ 0 := by
8    -- Reduce to a strict inequality in ℝ, then finish by linear arithmetic.
9    suffices h : (m : ℝ) < (n + 1 : ℝ) by
10     linarith
11   -- Move the goal back to ℕ.
12   norm_cast
13   -- Discharge the ℕ inequality from n > m.
14   omega
```

## D.2. Qualitative examples of WZ-Prover outputs

We conclude this section with representative WZ-Prover outputs, illustrating the two complementary roles played by the trained prover in our framework.

**Prompting protocol.** WZ-Prover is queried with a fixed instruction-following prompt that asks the model to (i) first write a brief proof plan, and (ii) then output a complete Lean 4 proof script. Concretely, we use the following prompt template, where `ts` is a placeholder for the current Lean proof state (the program replaces `ts` with the target theorem statement and its surrounding Lean context before inference):

**Prompt template (`ts` is replaced at runtime).**
Complete the following Lean 4 code:
"'lean4 ts"'
Before producing the Lean 4 code to formally prove the given theorem, provide a detailed proof plan outlining the main proof steps and strategies. The plan should highlight key ideas, intermediate lemmas, and proof structures that will guide the construction of the final formal proof.

Using this prompt, we show qualitative examples of the resulting model outputs. These examples illustrate the two complementary roles played by the trained prover in our framework.

**Two output modes.**

- **(A) Sketch-induced lemmas (symbolic-guided).** When the symbolic backend succeeds, a WZ sketch decomposes the original identity into many short Lean-checkable obligations. Among the most frequent are: *(i) term-ratio recurrences* (used to verify the WZ invariant and telescoping), and *(ii) non-vanishing denominator goals* (well-definedness conditions required by field_simp and normalization).

- **(B) Direct end-to-end proofs (symbolic-free).** For WZ-inapplicable identities or when decomposition fails, WZ-Prover attempts to prove the identity directly in Lean, without relying on the symbolic sketch.

**(A) Example: a sketch-induced lemma.** The following are typical lemma forms produced by symbolic decomposition. They are short, local, and appear repeatedly across WZ-style proof chains.

**(A1) Term-ratio recurrence (for simplification).**

```
1  import Mathlib
2  open Real Nat Finset BigOperators
3
4  set_option maxHeartbeats 8000000000
5  theorem rwz (n x : ℕ) (hby_cases : x ≥ n) :
6      ((-1) ^ (n+1) : ℝ) = ((-1) ^ n) * (-1) := by
7    -- typical local rewrite lemma used in ratio simplification
8    simpa [pow_succ] using (pow_succ (-1 : ℝ) n)
```

**(A2) Non-vanishing denominator (for `field_simp`).**

```
1  import Mathlib
2  open Real Nat Finset BigOperators
3
4  set_option maxHeartbeats 8000000000
5  theorem hwz (n m : ℕ) (h : n > m) : ((↑n - ↑m + 1) : ℝ) ≠ 0 := by
6    -- typical well-definedness obligation: denominator minimal factor ≠ 0
7    have : (0 : ℝ) < (↑n - ↑m + 1) := by
8      have : (m : ℝ) < (n + 1 : ℝ) := by
9        norm_cast; omega
10     linarith
11   linarith
```

**(B) Example: a direct identity proof (symbolic-free).** In contrast, the next example is an end-to-end identity proof generated by WZ-Prover without invoking WZ decomposition. It showcases a "global" proof that relies on Mathlib's analytic lemmas rather than sketch-induced local obligations.

```
 1  import Mathlib
 2  open Nat BigOperators
 3
 4  theorem idt_149 (n : ℕ) {x : ℝ} (hx : 0 ≤ x ∧ x < 1) :
 5      (1 - x) ^ (-(n + 1 : ℕ) : ℝ) = Σ' k : ℕ, (n + k).choose k * x ^ k := by
 6    have hx_abs : |x| < 1 := by
 7      have : -1 < x ∧ x < 1 := by
 8        constructor
 9        · linarith [hx.1]
10        · exact hx.2
11      simpa [abs_lt] using this
12
13    rw [neg_eq_neg_one_mul, Real.rpow_mul, Real.rpow_neg_one, inv_eq_one_div]
14    norm_cast
15    rw [Real.rpow_natCast, one_pow, div_eq_mul_inv]
16    -- main analytic lemma (name may vary slightly across Mathlib versions)
17    simpa using (tsum_choose_mul_geometric_of_abs_lt_1 (n := n) hx_abs)
```

Together, (A) and (B) reflect the intended complementarity: symbolic decomposition turns long identities into many short obligations that WZ-Prover can discharge reliably, while direct proving covers WZ-inapplicable identities without requiring a symbolic sketch.

## E. More Experimental Details

This appendix reports implementation and reproducibility details, covering additional experimental settings (tooling, token budgets, and optimization hyperparameters).

### E.1. Software Stack and Tooling

**Lean environment.** All proofs are checked in `leanprover/lean4:v4.25.0`. Kernel verification is treated as the sole correctness criterion.

**Symbolic backend.** We use SageMath `10.7` for symbolic computation, including WZ certificate/recurrence synthesis and auxiliary ratio simplifications. Sage is invoked programmatically via Python.

**Inference runtime.** Model inference is run with vLLM `0.13.0`. For baseline provers that rely on long chain-of-thought (Kimina-Prover, MA-LoT, DeepSeek-Prover-V2, and Goedel-Prover-V2), we cap the reasoning length at **16,384** tokens to reduce compute while keeping the sampling budget comparable (pass@32 where applicable). Other baselines follow the official decoding and token-limit settings provided in their released code.

### E.2. Evaluation Budget and Token Limits.

We report end-to-end success under pass@32 where applicable. To keep compute manageable for long-reasoning baselines (Kimina-Prover, MA-LoT, DeepSeek-Prover-V2, Goedel-Prover-V2), we cap the reasoning length at **16,384** tokens. Other baselines follow their official evaluation settings.

### E.3. Prompting Protocol

We use the same instruction-following structure throughout training and evaluation. Each instance asks the model to (i) provide a short proof plan that identifies key lemmas/tactics, and (ii) output a complete Lean4 proof script that closes the goal without using `sorry`. For WZ-induced obligations, the prompt additionally includes CAS-produced artifacts (e.g., rational certificates or simplified ratios) as explicit context, so that Lean-side verification focuses on well-definedness, algebraic rewriting, and recurrence/telescoping checks.

### E.4. Inference Cost of the Symbolic Pipeline

We measure the inference-time cost of WZ-LLM on seven LCI-Test instances where both direct proving and WZ-guided proving are applicable under pass@32.

*Table 7.* **Average inference cost on seven overlapping LCI-Test instances.** The symbolic front-end overhead is small compared with neural subgoal proving.

| Component | Avg. time (s) |
|---|---|
| WZ certificate generation | 0.17 |
| WZ-to-Lean sketch generation | 123.74 |
| Neural subgoal proving | 1530.16 |
| Complete symbolic pipeline | 1640.17 |
| Direct LLM proving baseline | 1639.38 |
| Avg. lemma obligations per instance | 27.43 |

Thus the added symbolic processing does not materially change the total inference cost in this setting; runtime is dominated by model sampling and Lean verification rather than certificate synthesis.

## F. Case Studies

We present two complementary case studies to illustrate how WZ-LLM operates in practice. Appendix F.1 focuses on the *symbolic proof sketch* produced by our framework, highlighting how symbolic computation induces a clear, executable global proof plan. Appendix F.2 then demonstrates how this sketch is instantiated into a complete end-to-end Lean4 proof through a single interactive command, showcasing the full automation pipeline of WZ-LLM (Algorithm 1).

### F.1. From an Automatically Generated Proof Sketch to Structured Obligations

This case study illustrates how WZ-LLM bridges symbolic proof planning and verifier-checked formal reasoning in Lean4. Given a target combinatorial identity, the framework first constructs a high-level *WZ proof sketch template* following the standard Wilf–Zeilberger reasoning pattern: normalization, recurrence/telescoping, and boundary-condition discharge. Rather than attempting unconstrained end-to-end proof search, this sketch explicitly exposes the global proof structure and reduces the original goal to a small set of well-scoped, Lean-checkable proof obligations. The sketch shown below is not a prompt template and is not generated by the LLM. It is the executable skeleton emitted by the symbolic computation component; WZ-Prover is invoked only for the marked Lean obligations.

The template below summarizes the canonical structure of a WZ-style proof sketch generated by WZ-LLM. Each step corresponds to a distinct class of obligations that can be independently verified in Lean, allowing scalable and modular formalization.

---

**WZ Proof Sketch Template in Lean 4**

**Annotation convention.** Subgoals delegated to WZ-Prover are marked as [LLM task: non-vanishing & ratio lemmas]. These obligations typically establish well-definedness conditions (nonzero denominators required by `field_simp`) and verify CAS-derived term-ratio identities used in recurrence checking.

```
theorem <IdentityName> (n : ℕ) (hn : <premises>) :
  (Σ k in Finset.range (n + 1), A n k) = B n := by
```

---

**Step 1: WZ certificate synthesis (CAS).**
Obtain a rational WZ certificate $R(n, k)$ from a symbolic backend (e.g., Sage).

```
let R : ℕ → ℕ → ℝ := WZ Certificate from CAS
```

---

**Step 2: Side conditions (well-definedness and boundaries).**
Prove non-vanishing conditions required for normalization and algebraic rewriting.

```
have ne_zeroA : ∀ n k, k < n → (A n k : ℝ) ≠ 0 := by
-- [LLM task: non-vanishing \& ratio lemmas]
have ne_zeroB : ∀ n, (B n : ℝ) ≠ 0 := by
-- [LLM task: non-vanishing \& ratio lemmas]
```

**Step 3: Normalization.** Reduce $\sum A(n,k) = B(n)$ to the canonical form $\sum F(n,k) = 1$ using a fixed lemma.

```
have WZ_aux (n : ℕ) (f : ℕ → ℕ → ℝ) (B : ℕ → ℝ)
(ne_zero : ∀ n, (B n : ℝ) ≠ 0) :
(Σ k in Finset.range (n+1), (f n k / B n : ℝ) = 1)
    ↔ (Σ k in Finset.range (n+1), f n k = B n) := by
-- fixed normalization lemma
```

**Step 4: CAS-assisted ratio lemmas.**
Precompute quotient identities to simplify recurrence checking in Lean.

```
have ratio_k : ∀ n k, k < n → A n (k+1)/A n k = <CAS_ratio_k> := by
-- [LLM task: non-vanishing \& ratio lemmas]
have ratio_n : ∀ n k, k < n → A (n+1) k/A n k = <CAS_ratio_n> := by
-- [LLM task: non-vanishing \& ratio lemmas]
have ratio_B : ∀ n, B (n+1)/B n = <CAS_ratio_B> := by
-- [LLM task: non-vanishing \& ratio lemmas]
```

**Step 5: WZ invariant.**
Verify the core WZ relation $F(n+1,k) - F(n,k) = G(n,k+1) - G(n,k)$.

```
let F := fun n k => A n k / B n
let G := fun n k => R n k * F n k
have WZ_invariant : ∀ n k, k < n →
F (n+1) k - F n k = G n (k+1) - G n k := by
-- [LLM task: non-vanishing \& ratio lemmas]
```

**Step 6: Telescoping and boundary discharge.**
Sum the invariant over $k$ to obtain $f(n+1) - f(n) = 0$.

```
let f := fun n => Σ k in Finset.range (n+1), F n k
have telescoping_step : ∀ n, f (n+1) - f n = 0 := by
-- [LLM task: non-vanishing \& ratio lemmas]
```

**Step 7: Constant extraction and unnormalization.**
Prove the base case, conclude $f(n) = 1$, and recover the original identity.

This template makes explicit which parts of the proof are handled symbolically, which reduce to algebraic side conditions, and which require Lean-level reasoning. Crucially, many boundary and non-vanishing conditions that are traditionally discharged manually in proof assistants are surfaced as explicit obligations and automatically handled by WZ-Prover.

## F.2. One-Command End-to-End Formal Proof in Lean

Beyond exposing a structured proof sketch, WZ-LLM is integrated into Lean as an interactive tactic. For a concrete identity such as

$$\sum_{k \in \text{range}(n+1)} (-1)^k \binom{n}{k} \frac{m}{m+k} = \frac{1}{\binom{m+n}{n}},$$

the user only needs to invoke a single command:

**wz_prove**

As illustrated in Figure 2, the Lean InfoView then automatically orchestrates the entire pipeline: it calls a symbolic engine (Sage) to synthesize WZ certificates and recurrence relations, decomposes the goal into structured subgoals, and interacts with WZ-Prover to generate Lean proof scripts for each obligation. Once the suggested proof is accepted in the InfoView,

*Figure 2.* Lean InfoView output of the **wz_prove** tactic. The tactic automatically invokes symbolic computation and the trained prover to generate Lean proof suggestions, which can be accepted to fill in the complete formal proof.

the complete proof is inserted into the Lean file, as shown in Listing 1, and verified by the kernel, yielding a fully checked end-to-end formal proof with minimal user intervention.

```
1  import Mathlib
2  open Real Nat Finset BigOperators Polynomial
```

```
3
4   set_option maxHeartbeats 8000000000
5   theorem MichaelZ_052 (n m : ℕ) {m : m ≠ 0} :
6       Σ k ∈ Finset.range (n + 1),
7           (-1 : ℝ) ^ k * (Nat.choose n k) * (m / (m + k))
8         = 1 / (Nat.choose (m + n) n : ℝ) := by
9
10      -----------------------------------------------------------------------
11      -- (0) WZ-style normalization: define summand A(n,k), target B(n),
12      --     normalized term F(n,k) := A(n,k)/B(n), and sum f(n):=ΣF(n,k).
13      --     Also define a certificate R(n,k) and its companion G(n,k)=R*F.
14      -----------------------------------------------------------------------
15      let A : ℕ → ℕ → ℝ := fun (n k : ℕ) =>
16          ((-1) ^ k * ↑(n.choose k) * (↑m / (↑m + ↑k)) : ℝ)
17      let B : ℕ → ℝ := fun n : ℕ => (1 / ↑((m + n).choose n) : ℝ)
18      let F : ℕ → ℕ → ℝ := fun n k => (A n k / B n)
19      let f : ℕ → ℝ := fun n => Σ k ∈ Finset.range (n+1), F n k
20
21      -- WZ certificate (precomputed by symbolic backend in the full system)
22      let R : ℕ → ℕ → ℝ := fun n k => (-(k + m)*k/((n - k + 1)*(n + 1)) : ℝ)
23      let G : ℕ → ℕ → ℝ:= fun (n k : ℕ) => R n k * F n k
24
25      -----------------------------------------------------------------------
26      -- (1) Non-vanishing side conditions: needed for field_simp/division.
27      --     These are typical well-definedness lemmas in WZ pipelines.
28      -----------------------------------------------------------------------
29      have ne_zeroA : ∀ n k : ℕ, k < n → (A n k : ℝ) ≠ 0 := by
30        intro n k htotalNumidx
31        simp [A]
32        -- choose n k ≠ 0 when k ≤ n (here k < n)
33        have h1 : ¬n.choose k = 0 := by
34          suffices h : 0 < (choose n k) by linarith
35          have h_le : k ≤ n := by omega
36          exact Nat.choose_pos h_le
37        -- m ≠ 0 (from implicit parameter), plus denominator m+k ≠ 0
38        have h2 : ¬m = 0 := by omega
39        have h3 : ¬(m : ℝ)+ (k : ℝ)= 0 := by norm_cast; omega
40        aesop
41      have ne_zeroB : ∀ n : ℕ, (B n : ℝ)  ≠ 0 := by
42        intro n
43        simp [B]
44        -- choose (m+n) n ≠ 0 since n ≤ m+n
45        have h1 : ¬(m + n).choose n = 0 := by
46          suffices h : 0 < (choose (m + n) n) by linarith
47          have h₁ : n ≤ m + n := by omega
48          exact Nat.choose_pos h₁
49        aesop
50      have ne_zeroB_succ : ∀ n : ℕ, (B (n + 1) : ℝ)  ≠ 0 := by
51        intro n
52        simp [B]
53        aesop
54      have ne_zeroB': ∀ n : ℕ, (B n : ℝ) * (B (n + 1) : ℝ)  ≠ 0 := by
55        intro m
56        rw [mul_ne_zero_iff]
57        exact And.intro (ne_zeroB m) (ne_zeroB (m + 1))
58      have ne_zeroAB' : ∀ n k : ℕ, k < n → (B n : ℝ) * (A n k : ℝ) ≠ 0 := by
59        intro n k hidx
60        rw [mul_ne_zero_iff]
61        exact And.intro (ne_zeroB n) (ne_zeroA n k hidx)
62
63      -----------------------------------------------------------------------
64      -- (2) Algebraic factoring helpers: isolate ratios like
65      --     A(n+1,k)/A(n,k) and B(n+1)/B(n) after field_simp.
66      --     These are bookkeeping lemmas that keep later steps readable.
67      -----------------------------------------------------------------------
```

```
68    have l₁ : ∀ n k : ℕ, k < n
69        → A (n + 1) k * B n / (B n * A n k) = A (n + 1) k / A n k := by
70      intro n k hidx
71      field_simp [ne_zeroA, ne_zeroB]
72
73    have l₂ : ∀ n k : ℕ, k < n
74        →  B (n + 1) * A n k / (B n * A n k) =  B (n + 1) / B n := by
75      intro n k hidx
76      field_simp [ne_zeroA, ne_zeroB]
77
78    have r₁ : ∀ n k : ℕ, k < n → B (n + 1) * (R n (k + 1) * A n (k + 1))
79        / (B n * A n k) = R n (k + 1) * (A n (k + 1) / A n k) * (B (n + 1) / B n)
     := by
80      intro n k hidx
81      field_simp [ne_zeroA, ne_zeroB]
82
83    have r₂ : ∀ n k : ℕ, k < n → B (n + 1) * (A n k * R n k)
84        / (B n * A n k) = R n k * (B (n + 1) / B n) := by
85      intro n k hidx
86      field_simp [ne_zeroA, ne_zeroB]
87
88    ------------------------------------------------------------------
89    -- (3) WZ normalization lemma: sum (A/B) = 1  ↔>  sum A = B.
90    --      This bridges the normalized constant-sum formulation back to
91    --      the original target identity at the very end.
92    ------------------------------------------------------------------
93    have WZ_aux (n : ℕ) (f : ℕ → ℕ → ℝ) (B : ℕ → ℝ)
94        (ne_zero : ∀ n : ℕ, (B n : ℝ) ≠ 0) :
95        Σ k ∈ Finset.range (n+1), (f n k / B n : ℝ) = (1 : ℝ)
96        ↔ Σ k ∈ Finset.range (n+1), f n k = B n := by
97      constructor
98      · intro h
99        rw [← Finset.sum_div, div_eq_iff (ne_zero n), one_mul] at h
100       norm_cast at h
101     · intro _
102       rw [← Finset.sum_div, div_eq_iff (ne_zero n), one_mul]
103       norm_cast
104
105   have Step1 := WZ_aux n A B ne_zeroB
106
107   ------------------------------------------------------------------
108   -- (4) Symbolic (ratio) facts used by the WZ identity:
109   --      aux₁: k-shift ratio    A(n,k+1)/A(n,k)
110   --      aux₂: n-shift ratio    A(n+1,k)/A(n,k)
111   --      aux₃: target ratio     B(n+1)/B(n)
112   --      These are exactly the kind of lemmas that a CAS-backed stage
113   --      tends to generate in bulk for many problems.
114   ------------------------------------------------------------------
115   have aux₁ (n k : ℕ) (htotalNumidx : k < n):
116       A n (k + 1) / A n k = -(n - k)*(k + m)/((k + m + 1)*(k + 1)) := by
117     simp only [A]
118     -- (-1)^(k+1) = (-1)^k * (-1)
119     have r1 : ((-1) ^ (k + 1) : ℝ) = ((-1) ^ k) * (-1) := by
120       rw [pow_succ]
121     rw [r1]
122     -- choose ratio: C(n,k+1) expressed from C(n,k)
123     have r2 : (↑(n.choose (k + 1)) : ℝ) = (↑(n.choose k)) * (n - k)/(k + 1) := by
124       have h₁ : n ≥ k := by omega
125       have h₂ : (k + 1 : ℝ) ≠ 0 := by linarith
126       field_simp
127       norm_cast
128       simp [Nat.choose_succ_right_eq]
129     rw [r2]
130     -- side conditions for field_simp
131     have h1 : (↑(n.choose k) : ℝ) ≠ 0 := by
```

```
132          norm_cast
133          suffices h : 0 < (n.choose k) by exact h.ne'
134          apply Nat.choose_pos
135          omega
136        have h2 : ((↑m + ↑(k + 1)) : ℝ) ≠ 0 := by norm_cast; omega
137        have h3 : ((↑m + ↑k) : ℝ) ≠ 0 := by norm_cast; omega
138        have h4 : (((-1) ^ k) : ℝ) ≠ 0 := by norm_num
139        have h5 : ((↑k + 1) : ℝ) ≠ 0 := by norm_cast
140        have h6 : (↑m : ℝ) ≠ 0 := by norm_cast
141        have h7 : ((↑k + ↑m + 1) : ℝ) ≠ 0 := by norm_cast
142      field_simp
143      grind
144
145      have aux₂ (n k : ℕ) (htotalNumidx : k < n):
146          A (n + 1) k / A n k = (n + 1)/(n − k + 1) := by
147        simp only [A]
148        -- choose lifting: C(n+1,k) expressed from C(n,k)
149        have r1 : (↑((n + 1).choose k) : ℝ) = (↑(n.choose k)) * (n + 1)/(n − k + 1)
      := by
150          have h₁ : n ≥ k := by omega
151          have h₂ : (↑n − ↑k + 1 : ℝ) ≠ 0 := by norm_cast
152          field_simp
153          norm_cast
154          simp [Nat.choose_mul_succ_eq]
155          omega
156        rw [r1]
157        -- side conditions for field_simp
158        have h1 : (↑(n.choose k) : ℝ) ≠ 0 := by
159          norm_cast
160          suffices h : 0 < (n.choose k) by exact h.ne'
161          apply Nat.choose_pos
162          omega
163        have h2 : ((↑m + ↑k) : ℝ) ≠ 0 := by norm_cast; omega
164        have h3 : (((-1) ^ k) : ℝ) ≠ 0 := by norm_num
165        have h4 : (↑m : ℝ) ≠ 0 := by norm_cast
166        have h5 : ((↑n − ↑k + 1) : ℝ) ≠ 0 := by
167          suffices 0 < (↑n − ↑k + 1 : ℝ) by linarith
168          simp only [sub_eq_add_neg, add_assoc]
169          norm_cast
170          omega
171      field_simp
172
173      have aux₃ : ∀ n : ℕ, B (n + 1) / B n = (n + 1)/(n + m + 1) := by
174        intro n
175        simp only [B]
176        -- ratio of binomial coefficients in B(n)=1/choose(m+n,n)
177        have r1 : (↑((m + (n + 1)).choose (n + 1)) : ℝ)
178            = (↑((m + n).choose n)) * (n + m + 1)/(n + 1) := by
179          have h₁ : (n + 1 : ℝ) ≠ 0 := by linarith
180          field_simp
181          norm_cast
182          have h₂ : (m + (n + 1)).choose (n + 1) * (n + 1)
183              = (m + n + 1).choose (n + 1) * (n + 1) := by
184            have h₂₁ : m + (n + 1) = m + n + 1 := by omega
185            rw [h₂₁]
186          have h₂' : (m + n).choose n * (n + m + 1)
187              = (m + n).choose n * (m + n + 1) := by
188            have h₂'₁ : n + m + 1 = m + n + 1 := by omega
189            rw [h₂'₁]
190          simp [h₂, h₂', Nat.choose_succ_right_eq, Nat.choose_mul_succ_eq]
191        rw [r1]
192        -- side conditions
193        have h1 : ((↑n + ↑m + 1) : ℝ) ≠ 0 := by norm_cast
194        have h2 : (1 : ℝ) ≠ 0 := by norm_num
195        have h3 : ((↑n + 1) : ℝ) ≠ 0 := by norm_cast
```

```
196      have h4 : (↑((m + n).choose n) : ℝ) ≠ 0 := by
197        norm_cast
198        push_neg
199        rw [← Nat.pos_iff_ne_zero]
200        apply Nat.choose_pos
201        linarith
202      field_simp
203
204    -- boundary helpers that appear in the telescoping boundary terms
205    have r_aux1 : ∀ n : ℕ, A (n + 1) (n+1) = ((-1)*(m + n) / (m + n + 1)) * A n n
    := by
206      intro n
207      simp only [A]
208      field_simp
209      have r1 : ((-1) ^ (n + 1) : ℝ) = ((-1) ^ n) * (-1) := by rw [pow_succ]
210      rw [r1]
211      norm_num
212      grind
213
214    have r_aux2 : ∀ n : ℕ, A (n + 1) n = (n + 1) * A n n := by
215      intro n
216      simp only [A]
217      field_simp
218      norm_num
219
220    -------------------------------------------------------------------
221    -- (5) Core WZ step: show f(n+1) - f(n) = 0, by telescoping:
222    --     F(n+1,k) - F(n,k) = G(n,k+1) - G(n,k)
223    --     Sum over k collapses to boundary terms, which cancel.
224    -------------------------------------------------------------------
225    have Step2 : ∀ n : ℕ, f (n + 1) - f n = 0 := by
226      intro n
227
228      -- pointwise WZ identity for each k (local recurrence + certificate)
229      have WZ (k : ℕ) (htotalNumidx:k < n) :
230          F (n + 1) k - F n k = G n (k + 1) - G n k := by
231        simp only [F, G]
232        -- normalize differences under common denominators
233        field_simp [ne_zeroA, ne_zeroB]
234        rw [← div_left_inj' (ne_zeroAB' n k (by omega))]
235        rw [sub_div, mul_sub, sub_div ]
236        -- split into ratios A(n+1,k)/A(n,k) and B(n+1)/B(n)
237        rw [l₁ n k (by omega), l₂ n k (by omega), r₁ n k (by omega), r₂ n k (by
    omega)]
238        -- insert the explicit symbolic ratios aux₁/aux₂/aux₃
239        rw [sub_eq_iff_eq_add, ← sub_mul]
240        nth_rw 2 [show B (n + 1) / B n = 1 * (B (n + 1) / B n) by grind]
241        rw [← add_mul]
242        simp [R]
243        rw [aux₁ n k (by linarith), aux₂ n k (by linarith), aux₃ n]
244        -- finish by algebraic simplification (denominators shown nonzero)
245        field_simp
246        have h1 : ((↑n - (↑k + 1) + 1) : ℝ) ≠ 0 := by
247          suffices h : k < (n : ℝ) by linarith
248          norm_cast
249        have h2 : ((↑k + 1) : ℝ) ≠ 0 := by norm_cast
250        have h3 : ((↑n - ↑k + 1) : ℝ) ≠ 0 := by
251          suffices h : k < (n + 1 : ℝ) by linarith
252          norm_cast
253          omega
254        have h4 : ((↑k + ↑m + 1) : ℝ) ≠ 0 := by norm_cast
255        have h5 : ((↑n + 1) : ℝ) ≠ 0 := by norm_cast
256        field_simp
257        ring
258
```

```
259      -- sum the pointwise WZ identity over k and telescope G(n,·)
260      calc f (n + 1) − f n
261        _ = (Σ k ∈ range (n+1), (F (n + 1) k − F n k)) + F (n + 1) (n+1) := by
262          simp [f]
263          rw [Finset.sum_range_add]
264          simp only [range_one, sum_singleton, add_zero, sub_add_eq_add_sub]
265        _ = (Σ k ∈ range n, (G n (k + 1) − G n k)) + F (n + 1) n − F n n + F (n +
    1) (n+1) := by
266          rw [Finset.sum_range_add]
267          simp only [range_one, sum_singleton, add_zero, add_left_inj, add_sub]
268          congr 2
269          apply Finset.sum_congr rfl
270          intro k hidx
271          simp only [mem_range] at hidx
272          exact WZ k hidx
273        _ = (G n n − G n 0) + F (n + 1) n − F n n + F (n + 1) (n+1) := by
274          -- telescoping: Σ (G(k+1)−G(k)) = G(n)−G(0)
275          congr 3
276          apply sum_range_sub
277
278      -- discharge boundary terms and show total difference is zero
279      simp [G, F]
280      field_simp [ne_zeroB n, ne_zeroB_succ n, ne_zeroB' n]
281      simp [mul_comm (B n)]
282      rw [add_sub_right_comm, ← sub_mul, add_assoc, ← add_mul]
283      nth_rw 1 [show B (n + 1) = (B (n + 1) / B n) * B n by field_simp [ne_zeroB]]
284      conv_lhs => enter[1];rw[← mul_assoc]
285      rw [← add_mul]
286      simp only [_root_.mul_eq_zero]
287      left
288      rw [sub_right_comm,← sub_one_mul,aux₃ n, r_aux1 n, r_aux2 n]
289      simp only [A, R]
290      field_simp
291      norm_num
292      grind
293
294      ----------------------------------------------------------------
295      -- (6) Conclude f(n) is constant (=1) by Step2 and base case f(0)=1.
296      ----------------------------------------------------------------
297      have Step3 : ∀ n : ℕ, f n = 1 := by
298        intro n
299        induction' n with n hm
300        · -- base case: f(0)=F(0,0)=A(0,0)/B(0)=1
301          simp [f, F, A, B]
302          have h1 : (↑m : ℝ) ≠ 0 := by norm_cast
303          field_simp
304        · -- inductive step: f(n+1)=f(n) from Step2, then use IH
305          exact (sub_eq_zero.1 $ Step2 n).trans (hm)
306
307      ----------------------------------------------------------------
308      -- (7) Unnormalize back to the original statement: Σ A = B.
309      ----------------------------------------------------------------
310      unfold A B at Step1
311      rw [Step1.1]
312      exact Step3 n
```

*Listing 1.* Complete Lean4 proof of the combinatorial identity $MichaelZ\_052$ generated by WZ-LLM.

