# OpenReview forum: "Automated Formal Proofs of Combinatorial Identities via Wilf–Zeilberger Guidance and LLMs"
_ICML.cc/2026/Conference — ICML 2026 spotlight_

### Official Review · Reviewer_5sEL · 2026-03-03

**Soundness:** 3
**Presentation:** 3
**Significance:** 2
**Originality:** 3
**Overall Recommendation:** 4
**Confidence:** 4

**Summary:**

The paper presents WZ-LLM, a neuro-symbolic framework that automates Lean 4 proofs for combinatorial identities by using the Wilf–Zeilberger method to decompose global goals into structured, machine-checkable subgoals. These subgoals are then discharged by WZ-Prover, a specialized model optimized through expert-verified bootstrapping and reinforcement learning. This approach achieves a 34% success rate on the LCI-Test benchmark, significantly outperforming general-purpose models like DeepSeek-V3.

**Compliance With Llm Reviewing Policy:**

Affirmed.

**Final Justification:**

The rebuttal has addressed my main concerns so I increased the score.

**Key Questions For Authors:**

1. Baseline Comparisons: The evaluation compares WZ-LLM against the 7B version of DeepSeek-Prover-V2 and the 685B DeepSeek-V3. Could the authors clarify why the full-scale DeepSeek-Prover-V2 (671B) or other state-of-the-art reasoning models specialized for formal proof were omitted? Evidence that these larger specialized models still fail without symbolic guidance would significantly strengthen the claim that the WZ-Sketch framework is necessary regardless of model scale.

2. Analysis of External Generalization: Table 3 shows a marginal 4% success rate improvement on CombiBench compared to the baseline. Is this limited gain primarily due to low applicability of the WZ method to that specific dataset, or the inability of WZ-Prover to solve the resulting subgoals? Distinguishing between decomposition failure and prover failure on external sets would help evaluate the framework's broader utility.

3. Symbolic-Only Baseline: The paper states that WZ-LLM proves 5 identities that the symbolic-only baseline fails on. Could the authors define exactly what tools or algorithms constitute this baseline and explain why it failed? For example, were these failures due to theoretical limits of creative telescoping or specific implementation gaps in SageMath?

4. Nature of Subgoal Failures: WZ-Prover achieves 73.34% accuracy on subgoals, leaving roughly 26% that still fail. Are these remaining failures due to mathematical complexity, such as difficult non-vanishing conditions, or hallucinations in Lean syntax? Understanding this bottleneck would clarify whether further improvement requires richer symbolic hints or enhanced neural reasoning.

**Limitations:**

While the paper presents a compelling neuro-symbolic approach to formalizing combinatorial identities, there are several areas where the work could be strengthened to better demonstrate its long-term impact and technical robustness. To further validate the framework's effectiveness and address current limitations, I offer the following suggestions for the authors:

*   **Expand symbolic coverage**: Currently, the framework is primarily limited to bivariate hypergeometric terms. Incorporating more advanced symbolic algorithms, such as q-analogues or multi-sum creative telescoping, would allow the system to handle a much broader range of identities found in modern combinatorics.
*   **Strengthen baseline comparisons**: While the 8B WZ-Prover is compared against DeepSeek-V3, the evaluation lacks a comparison against full-scale, 600B+ state-of-the-art reasoning models specialized for formal proof (e.g., DeepSeek-Prover-V2 671B). Including these would help determine if symbolic scaffolding remains essential as raw neural reasoning power scales up.
*   **Conduct detailed subgoal error analysis**: Since roughly 26.6% of sketch-induced subgoals still fail to be discharged by WZ-Prover, a qualitative analysis is needed to distinguish between failures caused by deep mathematical complexity and those caused by Lean-specific syntax hurdles or "hallucinations".
*   **Improve generalization on heterogeneous data**: The performance gain on external benchmarks like CombiBench is relatively modest compared to the in-domain LCI-Test. Refining the framework to better handle "non-standard" identities that do not strictly fit the canonical WZ-summation form would demonstrate greater practical utility.
*   **Automate routine formalization tasks**: The generation of non-vanishing denominator side-conditions remains a significant hurdle in formal verification. Developing specialized, automated Lean tactics to handle these routine algebraic proofs could further reduce the burden on the neural prover and streamline the end-to-end pipeline.

**Strengths And Weaknesses:**

**Strengths**
*   **Structured Decomposition**: Uses the Wilf–Zeilberger (WZ) method to break down long-horizon combinatorial proofs into localized, machine-checkable subgoals, effectively solving the "search explosion" problem in Lean 4.
*   **Effective Specialization**: Demonstrates that a symbolically-guided 8B model can significantly outperform massive general-purpose models like DeepSeek-V3 (34% vs. 1% success rate) on specialized benchmarks.
*   **Rigorous Pipeline**: Combines expert-verified bootstrapping with DAPO reinforcement learning to ensure the prover can handle the tedious algebraic details (e.g., non-vanishing side conditions) required for kernel-level verification.

**Weaknesses**
*   **Marginal Gains on External Data**: As shown in Table 3, the advantage narrows significantly on benchmarks like CombiBench, where the 16% success rate is only a 4% improvement over the 8B Goedel-Prover-V2 baseline.
*   **Incomplete Baseline Comparison**: The evaluation compares against a 7B version of DeepSeek-Prover-V2 but omits the full-scale 671B state-of-the-art version. This makes it difficult to determine if the symbolic framework is more effective than raw scaling in high-parameter reasoning models.
*   **Domain Dependency**: The performance gain is heavily concentrated on identities within the symbolic coverage of the WZ method; success rates drop sharply when such structural guidance is unavailable.
*   **Benchmark Scale**: The primary evaluation (LCI-Test) consists of only 100 problems, which may not be sufficient to establish the framework's broader robustness across the diverse field of combinatorics.

---

> ### Author Rebuttal · Authors · 2026-03-31
>
> We sincerely thank the reviewer for the helpful feedback and address the concerns in detail below:
>
> **Q1. Baseline Comparisons**
>
> To ensure fairness and reproducibility, Section 5.2 of this paper primarily evaluates a range of mainstream baseline models at the 7B–8B scale, and includes comparisons with general-purpose LLMs accessible via public APIs.
> Due to computational constraints, DeepSeek-Prover-V2-671B was not included. Instead, we report Gemini-3.1-Pro-Preview pass@32 results:
> |Model|Pass@32|
> |:-|-:|
> |Gemini-3.1-Pro-Preview|16|
> |WZ-LLM|34|
>
> This suggests that the structured symbolic decomposition provided by WZ-Sketch plays a crucial role in combinatorial identity proof tasks.
> We will include this comparison in the revised version.
>
> **Q2. Analysis of External Generalization**
>
> We agree that,  the improvements on CombiBench and PutnamBench-Comb are relatively limited.
> Further analysis suggests that the limited gain stems from the low applicability of WZ-style decomposition on these external benchmarks, rather than systematic failures of WZ-Prover on the resulting subgoals. Our framework is designed for automatic proving of combinatorial identities, whereas CombiBench and PutnamBench-Comb contain more general combinatorics problems.
>
> **Q3. Symbolic-Only Baseline**
>
> In this work, the “symbolic-only baseline” refers to instances where SageMath fails to generate a WZ certificate. We will clarify this definition in the revised manuscript.
>
> Failures of such symbolic methods generally arise from two factors. First, computational limitations: when the numerator and denominator degrees of the target WZ mate are high, the resulting linear system grows rapidly, leading to timeouts or no output. Second, theoretical limitations: the standard WZ method mainly applies to identities expressible as sums of hypergeometric terms; if an identity does not fall into this form, certificate generation will fail.
>
> For the five examples mentioned in the paper, our manual inspection indicates that the failures are due to the theoretical limitations. These identities involve non-hypergeometric terms or lack summation forms.
>
> **Q4. Nature of remaining subgoal failures**
>
> Based on our preliminary attribution analysis over 1,178 lemma subgoals, the current failures are not uniformly distributed but are instead concentrated in several mathematically more complex categories, as summarized in the table below. In particular, singularity (identity) lemmas exhibit the highest failure rate (73.68%), followed by recurrence relations (56.49%) and boundary conditions (48.91%).
>
> In contrast, simpler subgoals—such as non-vanishing denominators of a linear-algebraic form—have a much lower failure rate (9.34%), while those involving combinatorial constructs (e.g., choose, factorials) show a higher rate (26.55%).
>
> |Lemma Category|#Generated|#Proven|Avg. Proof Length|#Failed|Failure Rate|
> |:--|--:|--:|--:|--:|--:|
> |Non-vanishing (linear)|632|573|**1.50**|59|9.34%|
> |Non-vanishing (combinatorial)|177|130|**1.59**|47|26.55%|
> |Recurr. relations|239|104|5.95|135|**56.49%**|
> |Bound. conditions|92|47|14.32|45|**48.91%**|
> |Singularities(identity form)|38|10|41.20|28|**73.68%**|
> |**Total**|**1178**|864|–|314|26.66%|
>
> The current statistics suggest that the bottleneck is primarily associated with mathematical complexity and the burden of complex formal reasoning, rather than widespread hallucinations at the level of Lean syntax.
>
> **Q5. Domain Dependency and Expand symbolic coverage**
>
> We agree that the current performance gains are concentrated on identities within the symbolic coverage of the WZ method, indicating a degree of domain dependency. We chose the WZ method due to its mature algorithmic foundation and broad applicability for combinatorial identities.
> We conducted additional analysis on unsolved cases and explored extending symbolic methods:
>
> | Algorithm Category | #Solved |
> | :-| -: |
> | q-hypergeometric | 3 |
> | Multi-sum Creative Telescoping | 5 |
> | Trigonometric Creative Telescoping| 5 |
>
> This highlights significant room for improving the symbolic front-end. A key future direction is to extend it while preserving the unified pipeline of symbolic decomposition, proof sketching, and neural formal proving.
>
> **Q6. Automate routine formalization tasks**
>
> This comment provides valuable guidance for improving our current work. Our empirical observations suggest that these lemmas are typically short and exhibit clear patterns and regularity.
>
> We have implemented preliminary automation for two categories of frequently occurring side conditions based on the evalTactic (← (tactic| ... )) module.
> We will continue to extend this component.
>
> **Q7. Benchmark scale**
>
>  We agree that LCI-Test is insufficient to establish robustness across the broader field of combinatorics.
> Our work focuses on a narrower subtask—formal automated proving of research-level combinatorial identities. Thus, LCI-Test is designed to evaluate performance on this specific task.

---

> > ### Author Rebuttal · Reviewer_5sEL · 2026-04-03
> >
> > I appreciate the authors' comprehensive rebuttal, which has partially addressed my initial concerns. I am particularly encouraged by the supplementary results involving more advanced symbolic front-ends—such as **q-hypergeometric** and **multi-sum creative telescoping**—which demonstrate the framework's potential for handling **research-level combinatorial identities**.
> >
> > To further assess the potential impact of this work, I would like to inquire about the authors' plans for the public release of their research artifacts. Specifically:
> >
> > *   Do the authors intend to **open-source the WZ-LLM framework** and the **Lean 4 interactive tactic** described in the paper?
> > *   Will the **LCI-training and LCI-Test datasets**, including the manually formalized seed identities and the thousands of generated formal proof scripts, be made available to the community?
> >
> > Given that a stated objective of this work is to support the construction of **reusable verified libraries**, clarifying these plans would significantly help in evaluating the long-term significance and reproducibility of the contribution.

---

> > > ### Author Response · Authors · 2026-04-04
> > >
> > > Thank you for the encouraging follow-up and for recognizing the potential of the framework with more advanced symbolic front-ends, as well as the importance of reproducibility and reusable verified libraries. Our current plan is to publicly release the main research artifacts: **the WZ-LLM framework**, **the LCI-Test benchmark**, and **the LCI-training data**, including the manually formalized seed identities and the generated formal proof scripts. We are currently organizing and cleaning these resources for public release.

---

### Official Review · Reviewer_R6D6 · 2026-03-05

**Soundness:** 3
**Presentation:** 3
**Significance:** 3
**Originality:** 4
**Overall Recommendation:** 5
**Confidence:** 4

**Summary:**

This paper tackles a notoriously tricky problem for LLM-based provers: handling the long-horizon proof planning required for combinatorial identities. Unconstrained search in these spaces usually explodes. To bypass this, the authors lean on the classical Wilf-Zeilberger (WZ) method to generate a structural proof plan, which they then compile into executable Lean 4 proof sketches. An LLM (WZ-Prover) is deployed to discharge the remaining machine-checkable subgoals. To support this, they formalize a seed dataset of 307 identities for SFT and introduce a new 100-problem benchmark (LCI-Test). The resulting pipeline achieves a 34% success rate, a significant jump over standard LLM baselines.

**Compliance With Llm Reviewing Policy:**

Affirmed.

**Key Questions For Authors:**

Could you provide a breakdown of the failure modes for the 66% of LCI-Test where WZ-LLM fell short? Was the primary bottleneck the symbolic engine failing to find a WZ certificate, or the LLM failing to discharge specific Lean subgoals (e.g., non-vanishing denominators)?

For identities outside the strict hypergeometric scope where WZ cannot apply, what was the actual success rate of the direct LLM fallback?

How computationally expensive is the WZ certificate generation and translation pipeline at inference time compared to direct sampling from your baselines (like DeepSeek-Prover)?

**Limitations:**

Yes. The authors are upfront about the theoretical limitations of the WZ method's coverage.

**Strengths And Weaknesses:**

The neuro-symbolic angle here is exactly what this corner of the field needs right now. Purely symbolic methods often spit out WZ certificates that are incredibly tedious to verify in Lean without a dedicated bridge. By treating the symbolic engine as a sketch generator and the LLM as the tactical solver, the authors neatly sidestep the search explosion we typically see. I also appreciate the methodological rigor: relying strictly on the Lean 4 kernel for verification means we don't have to worry about false positives in the reported success rates. The LCI-Test benchmark is a solid contribution to the formal math community.
Weaknesses:
My primary hesitation revolves around the scope of the method and the depth of the error analysis. The WZ method is inherently restricted to terminating hypergeometric series. The authors mention that the LLM can fall back to direct proving when WZ fails, but we don't get a clear picture of how often this actually works in out-of-distribution scenarios.
Furthermore, while 34% is a strong relative improvement over the baselines, it still means the pipeline fails on roughly two-thirds of the benchmark. I wanted to see a much more granular breakdown of these failures. Is the LLM stumbling on the arithmetic side conditions, or is it failing to close the inductive steps?
Finally, the actual translation mechanism from a WZ certificate into a Lean proof sketch feels underexplored in the main text. The exact structural mapping is the core of the paper, but it gets a bit buried; I'd like to see the rational-function-to-Lean translation formalized more explicitly before we hit the appendix.

---

> ### Author Rebuttal · Authors · 2026-03-31
>
> We sincerely thank the reviewer for the careful reading of our work and for the positive evaluation of our method in both its design and empirical validation. Our detailed responses are provided below:
>
> **Q1.Failure Modes Breakdown for LCI-Test**
>
> We manually analyzed the 66 unsuccessful cases and will include this breakdown in the revision.
>
> The 66 failures can be categorized into two groups:
> 1. WZ proof sketch successfully generated, but some Lean subgoals remain unsolved (17 cases);
> 2. The symbolic stage fails to produce a usable WZ-style proof framework, and the direct LLM fallback also fails (49 cases).
>
> This indicates that the primary bottleneck for the remaining failures lies in the limited coverage of the symbolic framework. When the symbolic stage succeeds, the difficulty shifts to discharging a few critical Lean subgoals.
>
> For **the first category (17 cases)**, we analyze the types of unsolved subgoals:
> |Lemma Category|#Generated|#Proven|Avg. Proof Length|#Failed|Failure Rate|
> |:--|--:|--:|--:|--:|--:|
> |Non-vanishing denom.(linear)|632|573|**1.50**|59|9.34%|
> |Non-vanishing denom.(combinatorial)|177|130|**1.59**|47|26.55%|
> |Recurrence relations|239|104|5.95|135|**56.49%**|
> |Boundary conditions|92|47|14.32|45|**48.91%**|
> |Singularities(identity form)|38|10|41.20|28|**73.68%**|
> |**Total**|**1178**|864|–|314|26.66%|
>
> - Non-vanishing denominators (linear-algebraic / combinatorial forms): relatively low failure rates (9.34% / 26.55%), with short proof lengths (1.50 / 1.59 lines on average), indicating strong regularity;
>
> - Recurrence, boundary, and singularity lemmas show much higher failure rates (up to 73.68%) and longer proofs, indicating higher reasoning complexity and forming the main bottleneck.
>
> For **the second category (49 cases)**,these largely fall outside the current WZ-style pipeline rather than failing due to implementation issues. We explored additional symbolic tools:
> - Creative telescoping for multi-sums and trigonometric identities (e.g., via HolonomicFunctions in Mathematica), which solves 10 additional cases;
> - q-hypergeometric methods (e.g., Maple q-series tools), covering 3 more cases;
>
> However, these methods differ substantially in both their proof targets and proof strategies, and currently lack a unified, easily formalizable representation—an important direction for future work.
>
> **Q2.Success Rate of Direct LLM Fallback**
>
> For identities outside the WZ scope, we use direct end-to-end LLM proving as a fallback.
> Overall, the model directly proves 12 problems in total; among the 54 covered instances, the direct LLM fallback succeeds on 5 cases(wz_uncovered in Table 1).
>
> This suggests that while LLMs have some capability to directly formalize combinatorial identities without symbolic guidance, the success rate remains limited. This highlights substantial room for improvement. To address this, we are exploring data augmentation strategies based on structural features of identities to enhance model performance on broader classes of problems.
>
> **Q3.Computational Cost of WZ Pipeline**
>
> We compare the inference time under pass@32 on 7 overlapping instances where both methods apply:
>
> |Test ID|Number of Lemmas| WZ Cert Generation Time(s)|WZ-Sketch Generation Time(s)|**Symbolic Pipeline Time(s)** |**Direct LLM Proving Time (s)**|
> |:-|-:|-:|-:|-:|-:|
> |Gould_1_83|21|0.14|122.05|**1684.64**|2599.37|
> |Gould_3_92|52|0.08|143.52|**1867.66**|3097.04|
> |Jihuai_E1_1|13|0.09|113.47|**927.09**|1335.67|
> |Jihuai_E1_4|18|0.30|116.00|**1432.09**|1790.84|
> |Jihuai_example_1|23|0.13|140.31|2437.02|**723.02**|
> |Jihuai_1_6|16|0.08|103.58|1057.61|**963.88**|
> |Gould_3_116|49|0.40|127.22|2075.10|**965.84**|
> |**Average**|**27.43**|**0.17**|**123.74**|**1640.17**|**1639.38**|
>
> The total inference cost of WZ-LLM consists of three components:
> 1. WZ certificate generation : 0.17 seconds per problem (on average);
> 2. Translation from certificate to Lean proof sketch: 123.74 seconds per problem;
> 3. Subgoal proving : the dominant cost, approximately 1530.16 seconds per problem. On average, WZ-LLM generates 27.43 lemmas per instance.
>
> The total runtime remains dominated by the subgoal proving phase. On the 7 overlapping instances, the symbolic pipeline averages 1640.17 seconds per problem, compared to 1639.38 seconds for the direct baseline, indicating comparable cost. Notably, it is faster on 4 of the 7 instances, suggesting that incorporating symbolic processing does not significantly increase overall inference cost. We will include this analysis in the revision.
>
> **Q4.Translation from WZ Certificate to Lean Proof Sketch**
>
> Thank you for this valuable feedback. We agree that the translation from WZ certificates to Lean proof sketches is a core technical contribution, and that the current presentation in the main text is insufficient, with details mainly deferred to Appendix E.1.
> In the revision, we will move a simplified end-to-end example into Section 4 of the main text.

---

> > ### Author Rebuttal · Reviewer_R6D6 · 2026-03-31
> >
> > The authors provided a high-quality, evidence-based response that directly addresses the weaknesses identified in my initial review. Their manual analysis of the 66 unsuccessful cases from the LCI-Test provides the granular breakdown of failures I requested, distinguishing between 17 cases where Lean subgoals remained unsolved and 49 cases where the symbolic stage failed to produce a usable framework. The inclusion of success metrics for the direct LLM fallback, noting 12 total problems proved directly and 5 specific cases where WZ was inapplicable, clarifies the framework's relative strengths. Furthermore, the detailed timing data comparing the symbolic pipeline (1640.17s) to direct LLM proving (1639.38s) effectively mitigates concerns regarding the computational overhead of the WZ-to-Lean translation. I am also satisfied with the authors' commitment to moving the formalization of the WZ-to-Lean translation into Section 4 of the main text, ensuring this core technical contribution is properly highlighted. I am maintaining my score of 5 (Accept) as the paper remains a technically solid and significant contribution to formal automated theorem proving in combinatorics

---

### Official Review · Reviewer_FaV5 · 2026-03-14

**Soundness:** 3
**Presentation:** 2
**Significance:** 2
**Originality:** 3
**Overall Recommendation:** 4
**Confidence:** 3

**Summary:**

The paper focuses on automating proofs of combinatorial identities, which is more difficult than previous automated theorem proving (ATP) tasks because proofs require longer reasoning chains and the search space can easily explode. To address this challenge, the authors incorporate a Wilf–Zeilberger method symbolic engine to decompose the final proof goal into multiple subgoals.

To train an LLM to solve these subgoals, the authors construct a new dataset through manual formalization and automatic expansion. Experiments show that the proposed proving pipeline can solve more problems across three datasets.

**Compliance With Llm Reviewing Policy:**

Affirmed.

**Key Questions For Authors:**

- See weakness
- Why does WZ-Sketch + Goedel-Prover-V2 not improve the performance of Goedel-Prover-V2, given that the sketch should decompose the proof and reduce the difficulty?

**Limitations:**

The paper does not include a section discussing limitations, and claims that no potential negative societal impact must be specifically highlighted.

**Strengths And Weaknesses:**

Strengths

* The paper incorporates a well-chosen symbolic engine for a more challenging proving task.
* The new training data could benefit other provers as well. The WZ-Prover improves performance on all three datasets, demonstrating the effectiveness of both the generated data and the training pipeline.


Weaknesses

* The improvements are limited. Large gains mainly appear on the LCI-test after training on the same dataset, while generalization to the other two external datasets is relatively small.
* The experiments do not clearly validate whether the subgoal decomposition by the WZ-engine improves performance. The experiments show that the WZ-engine+other proves do not yeild improvements. The authors should provide statistics about the generated subgoals and analyze whether these subgoals increase the pass rate for both the baseline and WZ-Prover models.
* The ATP pipeline is difficult to follow. For example, it is unclear how the informal problem statement is translated into the executable WZ proof sketch. Is the input problem formalized and can be directly used by the prover? It is also unclear whether the WZ Proof Sketch Template in Appendix E.1 is a prompt template or an example output. A full example (input problem → prompt → WZ-engine inputs/outputs → final Lean4 proof code) would help clarify the entire process.

---

> ### Author Rebuttal · Authors · 2026-03-31
>
> We thank the reviewer for the insightful feedback and for recognizing the strengths of our symbolic engine and the empirical improvements across datasets.
> We now address the comments in detail:
>
> **Q1:Generalization on external datasets**
>
> We agree that, compared to the in-domain LCI-Test, our method achieves only modest gains on the general combinatorics benchmarks, CombiBench and PutnamBench-Comb.
>
> One key reason for this phenomenon is the mismatch in task distributions. Our framework is specifically designed for the automated proving of combinatorial identities, with its core mechanism based on WZ-guided goal decomposition and automated subgoal proving. In contrast, these benchmarks focus on more general combinatorial problems.
>
> As shown in Section 5.3, identity-type problems that are most closely aligned with our target setting account for only 3 and 1 instances in these benchmarks, respectively, and our method successfully solves them. This indicates that WZ-LLM retains some transferability to closely aligned subtasks.
> To achieve more significant improvements in broader combinatorial benchmark tests, future work will need to expand the model’s coverage of general combinatorial reasoning distributions; for example, by incorporating a broader range of combinatorial training data.
>
> **Q2:Effect of WZ-based subgoal decomposition**
>
> Thank you for your insightful observation; in fact, we also noted this phenomenon in Section 5.2.
> We would like to clarify that decomposing sub-goals does not necessarily lead to a direct improvement in end-to-end success rates. In our framework, the WZ engine decomposes the original problem into multiple necessary proof obligations that must all be satisfied, resulting in an 'all-or-nothing” characteristic.  A problem is counted as successfully **solved** only when all its associated subgoals are proven.
>
> To quantify this phenomenon, we further analyze the 46 problems reported in Table 2 of Section 5.2. Across these problems, the WZ engine generates a total of 1,178 lemma-level subgoals, with an average of 25.6 per problem (ranging from 8 to 52). The statistics are summarized as follows:
>
> |Model|#Proved|True Ratio|6≤False≤10|1≤False≤5|#solved|
> |-|-:|-:|-:|-:|-:|
> |WZ-Prover|864/1178|0.7334|5|31|29|
> |Goedel-Prover-V2|564/1178|0.4788|4|4|0|
>
> These results suggest that Goedel-Prover-V2 is incapable of leveraging WZ-based decomposition. At the lemma level, it successfully proves 564 out of 1,178 subgoals  with 4 problems having 1–5 subgoals remaining unsolved. This suggests that the model is generally close to end-to-end success but is limited by the absence of a few key lemmas.
>
> In contrast, after targeted training, WZ-Prover improves the lemma-level success rate on the same subgoals to 73.34% (864/1,178), and significantly reduces the number of remaining unsolved subgoals per problem. As a result, more instances transition from having only a few unresolved lemmas to being fully proven, leading to 29 out of 46 problems solved end-to-end.
>
> These findings suggest that the issue lies not in the invalidity of the WZ decomposition itself, but rather in the prover’s coverage of key WZ-style subgoals.
>
> **Q3:ATP pipeline is unclear**
>
> We thank the reviewer for the valuable suggestion. We provide the following clarifications:
>
> - Input format. The input to our prover is not natural language problem statements but formally specified combinatorial identities in Lean 4, which can be directly consumed by the prover (see Section 4.1, Line 203).
> - Source of the “WZ Proof Sketch Template.” The “WZ Proof Sketch Template” in Appendix E.1 is neither a prompt template nor generated by an LLM. Instead, it is an executable proof sketch automatically produced by our symbolic computation framework. Its role is to translate the proof plan derived from the WZ method into a structured Lean 4 proof skeleton, which is then completed by the prover through subgoal-level reasoning. Appendix E.2 provides a corresponding example output along with the final generated complete Lean code.
>
> We will revise the presentation in Appendix E.1 to make its origin and role more explicit. In addition, we will extract a concise end-to-end example from Appendix E.2 and include it in Section 4 to provide a more intuitive illustration of the overall pipeline.
>
> **Q4. The paper does not include a section discussing limitations.**
>
> We will include a dedicated Limitations section in the revised version. The main limitations are:
> - Limited applicability. As discussed in Section 3, our framework relies on WZ templates and is therefore mainly applicable to identities that can be reduced to WZ-eligible forms.
> - Dependence on training data. As shown in Section 5.2, the performance is sensitive to the constructed training data, which limits generalization to out-of-distribution settings.
> - Reliance on symbolic computation tools. The method depends on external tools to generate intermediate certificates, which may fail on complex problems.

---

### Decision · Program_Chairs · 2026-04-30

**Decision:**

Accept (spotlight)

**Comment:**

This paper focuses on automating proofs of combinatorial identities and the paper focuses on incorporating Lean into the pipeline. All the reviewers appreciated the contribution and think high of the paper.

I encourage authors to take all the feedback into account while preparing the final version.